# Addressing critiques refines global estimates of reforestation potential for climate change mitigation

Kurt A. Fesenmyer [1] ✉, Erin E. Poor [2], Drew E. Terasaki Hart[3,4], Joseph W. Veldman[5], Forrest Fleischman [6], Pooja Choksi [6], Sally Archibald [7], Mohammed Armani [8], Matthew E. Fagan [9], Evan C. Fricke[10], César Terrer [10], Natalia Hasler[11], Christopher A. Williams [12], Peter W. Ellis [13] & Susan C. Cook-Patton [3] ✉

Reforestation is a prominent climate change mitigation strategy, but available global maps of reforestation potential are widely criticized and highly variable, which limits their ability to provide robust estimates of both the locations and total area of opportunity. Here we develop global maps that address common critiques, build on a review of 89 reforestation maps created at multiple scales, and present eight reforestation scenarios with varying objectives, including providing ecosystem services, minimizing social conflicts, and delivering government policies. Across scenarios, we find up to 195 Mha (million hectares) are available (2225 TgCO$_2$e (teragrams of carbon dioxide equivalent) per year total net mitigation potential), which is 71–92% smaller than previous estimates because of conservative modeling choices, incorporation of safeguards, and use of recent, high-resolution datasets. This area drops as low as 6 Mha (53 TgCO$_2$e per year total net mitigation potential) if only statutorily protected areas are targeted. Few locations simultaneously achieve multiple objectives, suggesting that a mix of lands and restoration motivations will be needed to capitalize on the many potential benefits of reforestation.

Natural climate solutions (NCS) are ecosystem stewardship actions that protect, manage, and restore natural and working lands to provide measurable climate change mitigation[1], and have garnered increasing international and policy recognition[2]. Reforestation—the restoration of forest cover via tree planting, direct seeding, or natural regrowth in places where forests are absent but naturally occur—is especially promising because it is the largest and most cost-effective option for carbon removal[1–4]. However, there remains high uncertainty and controversy around where reforestation can be deployed.

Numerous global maps have been produced to identify areas where reforestation for climate change mitigation could technically and/or optimally occur[1,5–8]. Five high-profile global maps have been subjects of much criticism[9–25]: the Atlas of Forest Landscape Restoration Opportunities (FLRO) by Laestadius et al.[6], the NCS

[1]Tackle Climate Change, The Nature Conservancy, Boise, ID, USA. [2]Tackle Climate Change, The Nature Conservancy, Denver, CO, USA. [3]Tackle Climate Change, The Nature Conservancy, Arlington, VA, USA. [4]CSIRO Environment, Brisbane, QLD, Australia. [5]Department of Ecology and Conservation Biology, Texas A&M University, College Station, TX, USA. [6]Department of Forest Resources, University of Minnesota, St. Paul, MN, USA. [7]School of Animal, Plant and Environmental Sciences, University of the Witwatersrand, Johannesburg, South Africa. [8]College of Agriculture and Natural Resources, Kwame Nkrumah University of Science and Technology, Kumasi, Ghana. [9]Department of Geography and Environmental Systems, University of Maryland, Baltimore, MD, USA. [10]Department of Civil and Environmental Engineering, Massachusetts Institute of Technology, Cambridge, MA, USA. [11]George Perkins Marsh Institute, Clark University, Worcester, MA, USA. [12]Graduate School of Geography, Clark University, Worcester, MA, USA. [13]Tackle Climate Change, The Nature Conservancy, Portland, ME, USA. ✉e-mail: kurt.fesenmyer@tnc.org; susan.cook-patton@tnc.org

reforestation map by Griscom et al.[1], the Global Tree Restoration Potential map by Bastin et al.[5], the Global Priority Areas for Restoration by Strassburg et al.[7], and the Global Potential for Increased Storage of Carbon map by Walker et al.[8] (hereafter FLRO map, Griscom map, Bastin map, Strassburg map, and Walker map). Common critiques center on (1) definitions: the assumptions that determine where reforestation technically could occur, (2) data: the limitations of the source data used for mapping, and (3) precautions: the failure to consider potential perverse outcomes. These critiques have led to vociferous arguments in the literature[13,18,19,25] and media, leaving decision makers uncertain about the magnitude of global reforestation opportunity.

Definition critiques focus on the criteria used to delimit where reforestation could occur. Many of these objections are raised against broad definitions of forest that include open woodlands and savannas, where increasing tree cover reduces biodiversity and compromises ecosystem services[9–11,19]. Other criticisms address ecological factors like fire and herbivory, which can naturally limit tree cover but are often overlooked when defining where forest could occur[9,11,13,16,22].

Data critiques often focus on the limitations of source datasets. For example, Fagan demonstrated that some maps over-estimate reforestation potential in arid biomes because land use/land cover (LULC) maps systematically under-estimate sparse tree cover, and thus erroneously target areas that already have sufficient tree cover[17]. Other maps use coarse-scale LULC products that poorly differentiate land uses within mixed land cover classes (e.g., cropland mosaic classes that include up to 50% natural vegetation)[20]. Finally, some reforestation maps use coarse biome designations which ignore heterogeneity within regions and potentially overlook viable reforestation areas[11].

Precautionary critiques highlight that reforestation can have perverse outcomes and object to the lack of safeguards or practical consideration of those outcomes in reforestation maps. Maps that include croplands as reforestation opportunities are criticized for not sufficiently considering local food security in some regions, changes in food demand due to diet shifts, and/or leakage (i.e., conversion of ecosystems elsewhere for agriculture)[20]. Still other maps are criticized for failing to exclude areas where reforestation would exacerbate, rather than reduce, global warming due to changes in albedo[9,13,21,26]. Maps are also criticized for ignoring the equity implications of mapping reforestation opportunity in areas stewarded by communities with relatively low incomes and education levels, poor food security and health outcomes, weak rule of law and land tenure, and/or high reliance on subsistence agriculture[14,15,18,24].

Despite these critiques, global reforestation maps are widely referenced and used in cases ranging from policy documents to scientific research. For example, Google Scholar identifies over 6000 total citations for the FLRO, Griscom, Bastin, Strassburg, and Walker maps, the Bastin map launched the Trillion Trees movement (1t.org), and the International Panel on Climate Change (IPCC) used the Griscom map to determine the maximum potential of reforestation as a climate solution[3]. These uses underscore the need to address critiques directly and improve reforestation maps using updated science and sustainability principles[27].

In this work, we create a suite of global maps that (1) build on the strengths of existing maps, (2) address key critiques, (3) incorporate the latest high-resolution global datasets, and (4) demonstrate how areas of opportunity vary depending on different value-based methodological decisions. Rather than producing a single area estimate or conducting a prioritization exercise, we aim to provide multiple estimates that incorporate additional considerations so that decision makers can evaluate how much area is feasible and desirable for their reforestation efforts, given their specific circumstances.

## Results

### Existing reforestation maps

To learn from and build on previous efforts to map reforestation opportunities, we conducted a review of the existing literature (see "Methods") and found 89 studies published between 2011 and 2022 (see "Data availability"). We identified seven global maps, including four original maps (FLRO, Bastin, Strassburg, and Walker maps), a modification of the FLRO map (the Griscom map), and two applications of the Griscom map (refs. 28,29). We identified ten regional maps (including two that use the FLRO map), which mainly covered tropical regions. Finally, we identified 15 national and 57 sub-national maps, which occur in 10% of all countries (i.e., 20 countries). Although some areas have high coverage (e.g., states in Brazil's Atlantic Forest ecoregion have at least 20 maps available), most countries must rely on global or regional products for estimating reforestation opportunity (Supplementary Fig. 1).

We found that reforestation maps are generally produced using a common process, regardless of extent (Fig. 1). First, forest is defined, typically using tree cover, canopy cover, or biomass thresholds. Next, the area where forest could occur is identified (hereafter, forest potential). For example, the FLRO, Bastin, and Walker maps use biophysical constraints such as climate or soil variables to model forest potential. In other cases, as with the Strassburg map, areas of LULC change (i.e., deforestation) are used to infer forest potential. Next, the maximum area where reforestation could occur within the forest potential area is identified (hereafter, maximum reforestation potential) by accounting for practical limitations to reforestation, such as existing forest. Many maps also apply safeguards to minimize perverse outcomes and further limit the area where reforestation could occur (hereafter, constrained reforestation potential). For example, croplands are commonly removed in this step because reforestation of these lands can affect food security[30] and/or result in leakage[31]. Some analyses then use additional factors to further evaluate trade-offs and benefits or to prioritize areas within the maximum or constrained reforestation potential (hereafter contextualized reforestation potential).

Across the 89 reforestation opportunity maps, however, there is no universal standard for how spatial data is used for mapping forest potential, maximum reforestation potential, or constrained reforestation potential. These maps use 28 different types of spatial data to map where forest or reforestation potential can occur (hereafter suitability factors) and 22 different types of spatial data to map where forest or restoration cannot occur (hereafter exclusions) (Fig. 2 and Supplementary Table 1). Deforested areas, croplands, and pasture are the most frequently used suitability factors (used in 20 or more maps) (Fig. 2). Urban areas and existing forest are the most commonly applied exclusions (used in more than a third of all maps), followed by water, croplands, wetlands, and barren lands (Fig. 2). Overlays used for mapping contextualized reforestation span the greatest diversity of factors (30 across existing maps) with neighborhood metrics (e.g., forest pattern and distance to forest), economic factors (e.g., opportunity costs of the non-forest land use), terrain factors (e.g., slope), and carbon sequestration most commonly used (Fig. 2 and Supplementary Table 1). Sometimes the same factor is used as a suitability criterion, as an exclusion, and/or as an overlay in the same analysis (Fig. 2). For example, the Strassburg map uses cropland and pasture within the forest potential to identify the maximum reforestation potential and uses cropland and pasture productivity and yield to further prioritize areas within the contextualized reforestation potential.[7]

We compared the spatial overlap and agreement of the four original global forest potential maps (FLRO, Bastin, Strassburg, and Walker; we excluded the three global analyses[1,29,32], which re-use these maps). Combining the areas identified as forest potential across the four maps results in 9455 Mha (million hectares), equivalent to 74% of global ice-free land area. Individual map estimates range from

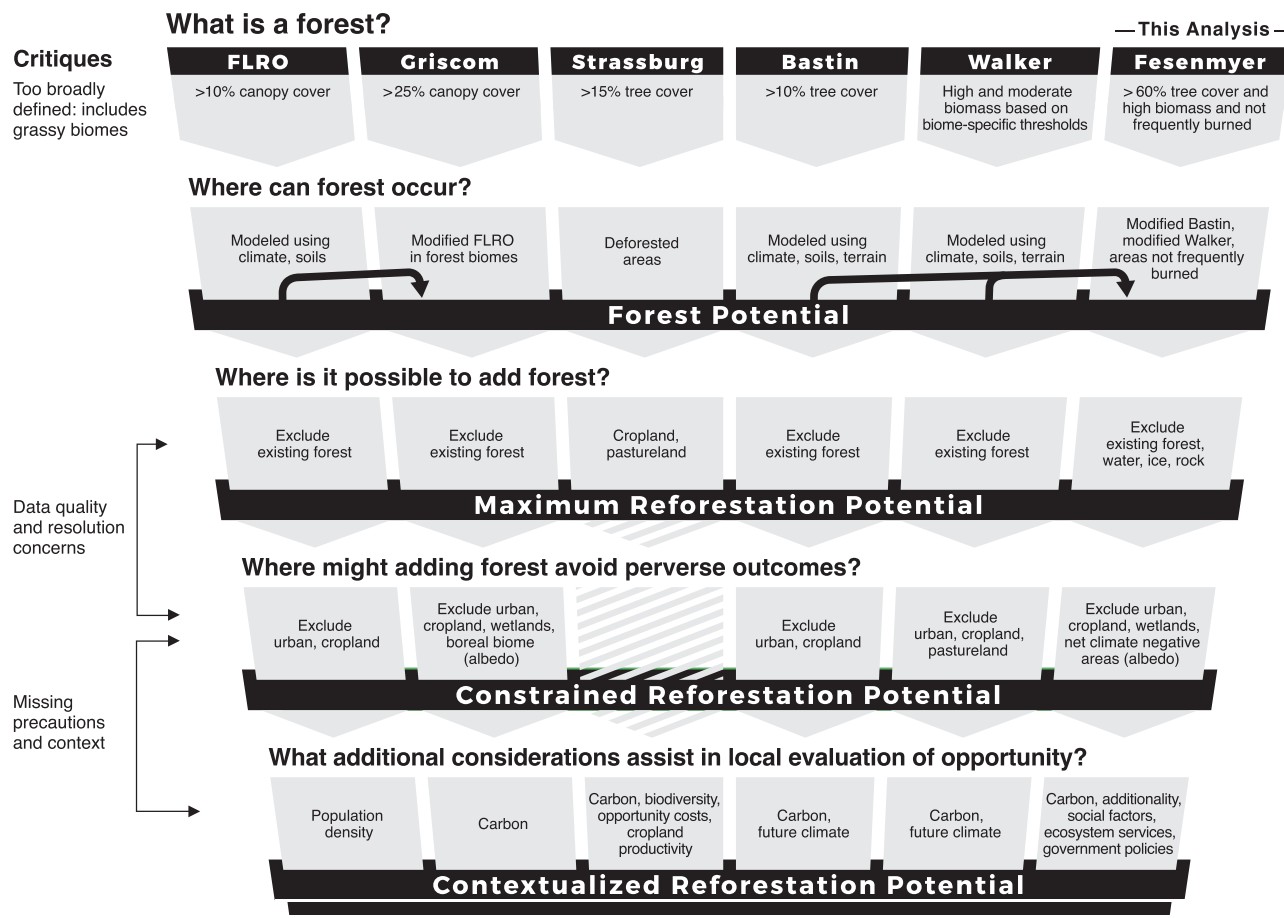

**Fig. 1 | Conceptual diagram of the process of mapping reforestation potential.** Reforestation mapping generally follows the following steps: (1) defining forest, (2) mapping forest potential (where forest can occur), (3) mapping reforestation potential within the potential forest, first as maximum reforestation potential, and then as constrained reforestation potential (maximum reforestation potential with safeguards), and (4) using overlays to provide additional information so that decision makers can evaluate the feasibility and desirability of the maximum or constrained reforestation potential given specific circumstances. Critiques from the literature associated with these steps are shown to the left. Details from unique existing global reforestation maps and this analysis are shown in columns.

6160 Mha (Strassburg) to 8682 Mha (Bastin) (Supplementary Table 2, see "Methods"). All four maps agree in 57% (5412 Mha) of their combined extent, but at least three maps agree in 77% (7246 Mha) (Supplementary Fig. 2). Thus, there is moderate convergence among products despite variation in how forest is defined: FLRO and the Bastin map use 10% tree cover threshold to define forests, the Strassburg map uses 15%, whereas the Walker map uses a potential biomass criterion (Fig. 1 and Supplementary Table 2).

Among the seven global analyses, five provide original maps of reforestation potential (FLRO, Griscom, Bastin, Strassburg, and Walker; we excluded one global analysis that reuses Griscom[28] and another that maps reforestation potential using future land use scenarios[29]). For identifying reforestation potential, the Strassburg map only identifies maximum reforestation potential (1537 Mha) (Supplementary Table 2). The remaining four global maps (FLRO, Griscom, Bastin, and Walker) go further and identify constrained reforestation potential (Supplementary Table 2). The combined extent of constrained reforestation potential for these four is 4288 Mha (34% of global ice-free land area) (see "Methods"). Constrained reforestation potential varies more dramatically than forest potential among products, with nearly a 3-fold difference in extent (678 Mha in the Griscom map to 2509 Mha in the FLRO map) (Supplementary Table 2). Further, all four maps share only 23 Mha in common (0.24% of combined forest potential area, and 0.9–2.7% of individual constrained reforestation maps), though at least three maps agree in 252 Mha (3% of combined forest potential area, and 10–30% of individual constrained

reforestation maps) (Supplementary Fig. 3). This striking variation is due to differences in forest potential mapping described above, as well as whether pastures are used as exclusions, and differences in the spatial resolution, time period, and LULC definitions in the datasets used to exclude existing forest, cropland, and/or pasture (Supplementary Table 2).

## Mapping forest potential

Given the issues with existing reforestation maps, we created a series of reforestation opportunity maps that addressed common critiques, beginning with improvements to mapping forest potential. First, we defined forests much more conservatively, as areas that can support 60% or more tree cover, to exclude open woodlands and savannas[33–35]. Second, because there is uncertainty associated with any one product, we used multiple lines of evidence to identify forest potential by combining the two global layers that mapped uncertainty around forest potential (i.e., the Bastin and Walker maps)[5,8]. We restricted forest potential to pixels where the Bastin map predicts at least 60% potential tree cover, the Walker map predicts closed forest, and both maps have low uncertainty. This resulted in a preliminary forest potential area of 2393 Mha. Finally, because the Bastin and Walker models overestimate forest potential in ecosystems with frequent fires[13], we eliminated areas with two or more non-cropland fires during 2002–2022, under the assumption that fire frequencies of at least two per decade can substantially limit tree density (see "Methods"). This reduced the preliminary forest potential area by 150 Mha (6.3%). These

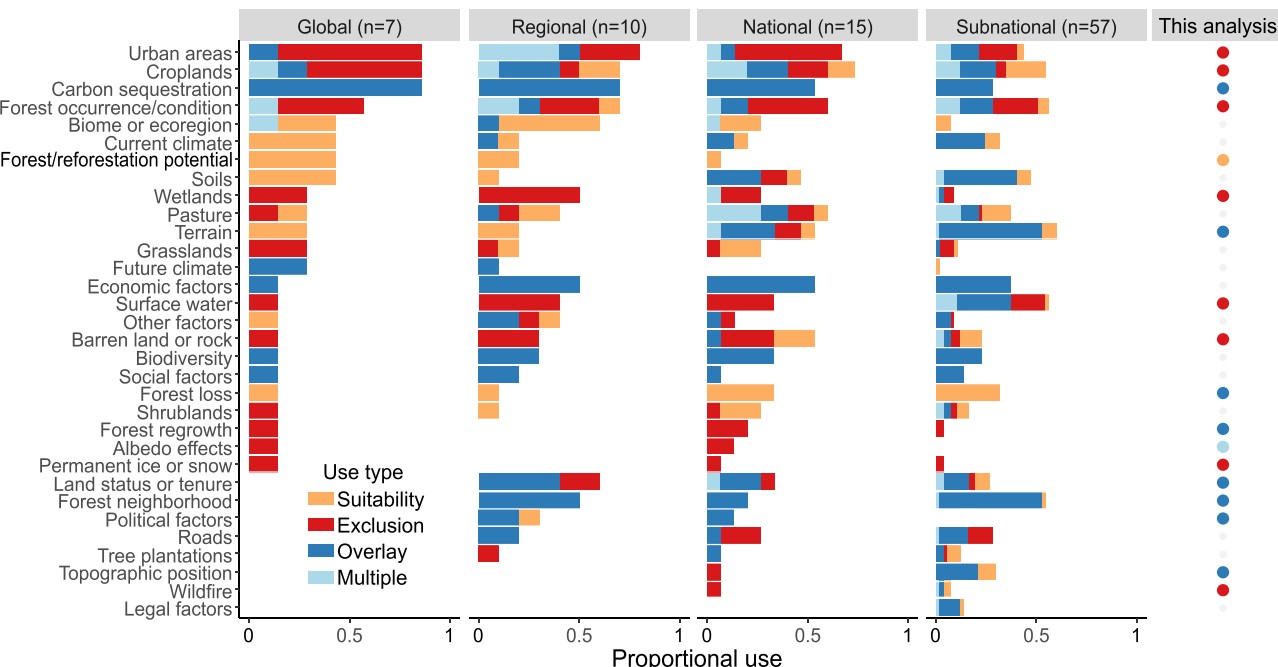

**Fig. 2 | Datasets used in reforestation maps.** Suitability, exclusion, and overlay datasets used by peer-reviewed reforestation maps that were published 2011–2022 (*n* = 89), and by this analysis. Datasets are partly sorted by prevalence in a given geography, top to bottom (sorting global first, then regional, then national, then subnational). Some datasets are used consistently across scales. For example, forest loss is always used to define suitability, and economic factors are always used as an overlay. However, other datasets are used in different ways depending on the individual product and scale of the analysis. For example, croplands are used as suitability, exclusion, and overlay factors in different products or within the same product.

three modifications narrowed forest potential to 2242 Mha (Supplementary Fig. 4 and Supplementary Table 2), or 26–36% of the area presented by the individual FLRO, Bastin, Strassburg, or Walker maps (Supplementary Table 2).

## Mapping maximum and constrained reforestation potential

To map maximum and constrained reforestation potential, we filtered our forest potential map to places where additional forests could grow given practical constraints and safeguards. We used four exclusions related to implementation limitations (existing forest, open water, bare ground, and permanent ice/snow) to identify maximum reforestation potential and five exclusions as safeguards (croplands, built-up lands, peatlands, wetlands, and albedo) to identify constrained reforestation potential (see "Methods", Supplementary Table 3). Except for permanent ice/snow and albedo, these exclusions are all commonly used in previously published maps (more than 12 uses). For all exclusions except peatlands and albedo, we used 10–30 m resolution global products that map individual LULC types (i.e., without mosaic LULC types) for 2019–2020[36–40] to address data critiques. Implementation exclusions remove 1937 Mha (86.4%) from the forest potential map, largely due to existing forest cover (1914 Mha or 85.3% of the forest potential area), resulting in a maximum reforestation potential of 305 Mha (Table 1). Precautionary exclusions remove an additional 110 Mha (4.9%). After we apply all exclusions, we identify 195 Mha of constrained reforestation potential (Fig. 3b and Table 1).

This area is 71–92% smaller than the constrained reforestation potential identified in other global products (Supplementary Table 2), and only intersects with other global products in 24–98 Mha (13–50% of the constrained potential in our map) (Supplementary Table 4). Our exclusions overlap with large areas (60–70%) identified as constrained reforestation potential in other products (Supplementary Table 4). In contrast, other global products missed substantial areas of our constrained reforestation potential because they used older or coarser resolution existing forest datasets with different criteria for forest

occurrence (132–153 Mha, or 67–78% of our constrained reforestation potential) or used older and coarser cropland and/or pasture maps as exclusions (50–59 Mha, or 26–30% of our constrained reforestation potential; Supplementary Table 4).

## Contextualized reforestation potential

We estimate that reforestation, if implemented in the constrained reforestation potential area, would deliver 2225 TgCO$_2$e (teragrams of carbon dioxide equivalent) per year of net climate benefits for the first 30 years of regrowth after deductions for albedo[26,28] (Table 1). The vast majority of this may represent additional climate benefit (i.e., would exceed business-as-usual forest recovery). When we calculate the ratio of forest loss to gain in one-degree cells using a 2000–2020 forest extent and change map[38] (Supplementary Fig. 5), and use that ratio to proportionally scale the area and mitigation estimates in each cell, we approximate that reforestation activities would likely be additional within 169 Mha (88.7%) of the constrained reforestation potential and achieve 2086 TgCO$_2$e per year (93.7%) of net climate benefit.

There are an estimated 98 million people who inhabit the constrained reforestation potential (Table 1). Because global maps can never fully capture the priorities of local communities, we did not attempt to prioritize areas for reforestation. Instead, we used additional overlays to help characterize what trade-offs and benefits might result within the constrained reforestation potential scenario. We clustered these into seven additional scenarios (Table 1) that subset the constrained reforestation potential based on practical considerations or on motivations for reforestation drawn from our review of existing maps and their critiques.

We created three avoiding social conflicts scenarios that sought to minimize the risk that reforestation results in injustices, displacement, and food insecurity for local communities. These three scenarios limit constrained reforestation potential to (1) countries where individuals and communities are likely able to influence decision-making through participatory or representative processes (high individual rights

## Table 1 | Reforestation potential and scenarios

| Name (and scenario group, where applicable) | Description and rationale | Global area (Mha) | Global net mitigation potential (TgCO₂e per year) | Population directly impacted (million) |
|---|---|---|---|---|
| Maximum reforestation potential | Forest potential area with removal of existing forest, open water, persistent barren land, and permanent ice/snow land cover. These exclusions account for limitations to the implementation of reforestation. | 305 | 3114 | 326 |
| 1. Constrained reforestation potential | Maximum reforestation potential with the removal of croplands, built-up lands, wetlands, peatlands, and areas with negative net climate benefits because of albedo change. These precautionary exclusions serve to avoid perverse outcomes from reforestation[26,30,31,71,72]. | 195 | 2225 | 98 |
| 2. High individual rights scenario (avoiding social conflicts) | Constrained reforestation potential is limited to countries where individuals and local communities have the ability to influence decision-making, as indicated by constitutionally protected civil liberties, strong rule of law, an independent judiciary, and effective checks and balances to government and/or majority power[14]. | 121 | 1474 | 49 |
| 3. Secure land tenure scenario (avoiding social conflicts) | Constrained reforestation potential limited to countries with secure land tenure, as indicated by a composite measure derived from multiple datasets to address missing data (see Supplementary Table 5). Formally recognized land rights and tenure have been identified as enabling conditions for successful reforestation[15]. | 116 | 1428 | 47 |
| 4. Low rural livelihood conflict scenario (avoiding social conflicts) | Constrained reforestation potential excluding areas within a 1-h walk from communities with high deprivation, where reforestation may conflict with the livelihoods of impoverished rural populations with high nature-dependence for food and fuel[14] and vulnerability to land use change[18,58]. | 158 | 1591 | 67 |
| 5. Forest neighborhood scenario (ecosystem services) | Constrained reforestation potential limited to areas with greater than 30% forest land cover within a 5 km radius. These locations have high biodiversity benefits and high potential for natural regeneration[41]. | 161 | 1777 | 77 |
| 6. Water quality scenario (ecosystem services) | Constrained reforestation potential limited to hillslopes of 20–35% or floodplains. These locations are a focus of local reforestation efforts (e.g., the Nature Conservancy/American Forest Foundation Mississippi Delta Floodplain Restoration Program) due to their low potential for land use change conflict, high water quality benefits (e.g., nutrient retention and soil stabilization[42]), and contribution to climate adaptation (e.g., flood mitigation[43,44]). | 71 | 814 | 39 |
| 7. Protected areas scenario (government policies) | Constrained reforestation potential limited to protected areas. Land management in these areas may emphasize protection and restoration, resulting in low land use conversion risk and greater carbon durability and permanence following restoration. | 6.3 | 53 | 1.7 |
| 8. National forest restoration goal scenario (government policies) | Constrained reforestation potential limited to countries that have committed to restoring or increasing forests through Land Degradation Neutrality, National Biodiversity Strategies and Action Plans, Nationally Determined Contributions (Paris Agreement), or Bonn Challenge pledges[48]. | 106 | 1665 | 57 |

Name, description, global area (in million hectares), global net mitigation potential (in teragrams carbon dioxide equivalent per year), and population directly impacted (in million people) in the maximum reforestation potential, constrained reforestation potential, and seven additional scenarios derived from the constrained reforestation potential. Supplementary Table 3 outlines the source datasets used to generate each scenario.

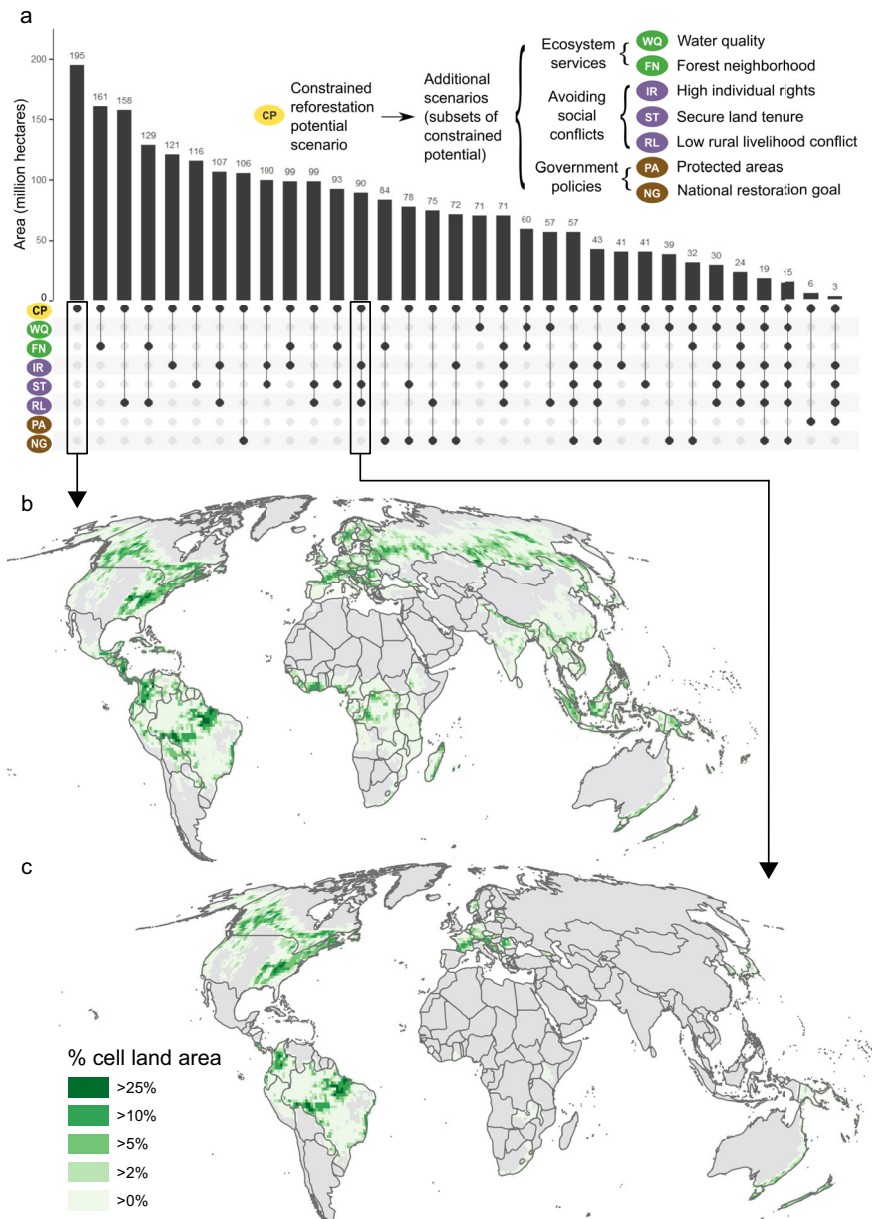

**Fig. 3 | Agreement (intersection) of reforestation scenarios and scenario combinations in this analysis, with example maps. a** Comparison of the area of seven scenarios to the constrained reforestation potential, and the area of agreement of 25 select scenario combinations (area shown in bars, identity of scenario combinations shown below the bars by circles), **b** percent land area in the constrained reforestation potential scenario by one-degree cells, and **c** percent land area where all avoiding social conflicts scenarios agree (intersect) within the constrained reforestation potential by one-degree cells. Panel **c** has less total area than panel **b** due to removal of areas with low individual rights, insecure land tenure, and/or potential conflict with nature-dependent livelihoods. Note that 102 other possible scenario combinations are not shown in **a**.

scenario, Supplementary Fig. 6a), (2) countries with indicators of land tenure (secure land tenure scenario, Supplementary Fig. 6b), and (3) areas without nature-dependent, vulnerable populations whose food and fuel needs may especially conflict with reforestation (low rural livelihood conflict scenario, Supplementary Fig. 6c). These scenarios total 121, 116, and 158 Mha respectively, potentially resulting in 1474, 1428, and 1591 TgCO$_2$e per year net climate mitigation and affecting 49, 47, and 67 million people (Table 1). The three scenarios avoiding social conflicts intersect in 90 Mha (Fig. 3a, c).

We created two scenarios focused on achieving ecosystem services other than carbon sequestration: one limited to areas with high nearby forest cover (Supplementary Fig. 7a) where biodiversity outcomes may be high[41] (161 Mha and 1777 TgCO$_2$e per year net climate

mitigation), and another limited to floodplain or moderate slope areas (Supplementary Fig. 7b) where water quality and climate adaptation benefits may be high[42–44] (71 Mha and 814 TgCO$_2$e per year net climate mitigation) (Table 1). These scenarios intersect in 60 Mha (Fig. 3a).

The final two scenarios limit the constrained reforestation potential based on government policies, either through protected areas (Supplementary Fig. 8a), where reforestation may be compatible with other management objectives and have greater durability, or forest restoration goals (Supplementary Fig. 8b), where dedicated funding, reforestation infrastructure such as tree nurseries, and other resources may incentivize reforestation. The protected area scenario has the smallest total area, mitigation opportunity, and population affected across all scenarios, at 6.3 Mha, 53 TgCO$_2$e per year net

climate mitigation, and 1.7 million people (Table 1). The national forest restoration goal scenario covers 106 Mha, 1665 TgCO2e per year net climate mitigation, and 57 million people (Table 1).

In general, we find it is hard to optimize for all factors. There are only 15 Mha where seven of the scenarios intersect (Fig. 3a) and 0.5 Mha where all eight intersect. However, multiple outcomes can be achieved. For example, the majority of the constrained reforestation potential (83%) occurs close to existing forest, where natural regeneration is likely and reforestation may enhance biodiversity[41] (Fig. 3a). Similarly, the low rural livelihood conflict scenario intersects with 81% of the constrained reforestation potential (Fig. 3a), indicating that it may be possible to reforest substantial areas without negatively impacting the livelihoods of those most dependent on nature. The co-occurrence of more than half of the constrained reforestation potential in countries with restoration goals suggests there is both ample opportunity and political systems that could be conducive to enable reforestation at a broad scale (Fig. 3a). Further, even though it is difficult to meet multiple conservation objectives simultaneously (Fig. 3a), we do find 57 Mha in countries with a restoration commitment and where restoration is less likely to cause conflicts, given secure land tenure, rights of individuals, and high standards of living. (Fig. 3a).

## Sensitivity tests

If we do not remove areas with frequent fire, forest potential and constrained reforestation potential increase by 7% and 24%, respectively (Supplementary Table 2 and Supplementary Fig. 9a), while if we use a 50% potential tree cover criterion (instead of 60%) to determine forest potential, then forest potential and constrained reforestation potential increase by 19% and 36% respectively (Supplementary Table 2 and Supplementary Fig. 9b). Using liberal criteria to define forest (>30% potential tree cover and lower biomass thresholds) and removing areas with frequent fire substantially increases forest potential (5095 Mha, an increase of 127%) and constrained reforestation potential (845 Mha, an increase of 333%; Supplementary Table 2 and Supplementary Fig. 9c). Using the same liberal definition of forest but without removing areas with frequent fire further increases forest potential (5788 Mha, an increase of 158%) and constrained reforestation potential (1135 Mha, an increase of 481%; Supplementary Table 2). Using alternative forest or cropland datasets as exclusions results in minor differences to the total constrained potential (1.8% and 6.6% larger area, respectively), but substantial differences in some geographies (Supplementary Tables 2 and 3 and Supplementary Fig. 10a, b). If we do not use croplands as an exclusion, constrained reforestation potential increases by 31% (to 255 Mha Supplementary Table 2 and Supplementary Fig. 10c). Finally, using different land tenure datasets in the secure land tenure scenario (Supplementary Table 5), results in 72–125 Mha of contextualized reforestation potential (62–108% of the primary land tenure scenario area; Supplementary Table 5). Thus, our results are most sensitive to the tree cover threshold used to define forest for forest potential mapping and maps of land tenure.

## Discussion

Global restoration mapping efforts, including reforestation maps, are used for estimating the overall magnitude of opportunity, supporting global policy development, and mobilizing resources to places with greater relative opportunity[45]. Prior attempts to map reforestation have faced multiple critiques[11,17,18,20], potentially limiting those uses. By addressing those critiques, our analysis shrinks the area of constrained reforestation potential in previous global estimates from between 678 Mha and 2509 to 195 Mha, a 71–92% reduction (Supplementary Table 2). Reforesting this latter area would result in 2225 TgCO2e per year of net climate mitigation for the first 30 years of regrowth, or roughly 5% of the sum of global fossil fuel and land use change

emissions in 2022, indicating that reforestation can still offer substantial climate change mitigation potential and remains the largest available carbon removal solution[1].

These results unsurprisingly fall at the lower range of previous constrained reforestation potential estimates (Supplementary Table 2), largely due to our use of a conservative forest definition (closed forest and at least 60% potential tree cover) that excludes savannas, and our choice to incorporate fire data into forest potential mapping. Indeed, only 5–24% of the constrained reforestation potential identified in other global maps intersects with our forest potential map (Supplementary Table 5). However, we map a more comparable constrained reforestation potential area to the FLRO, Griscom, Bastin, and Walker maps (1135 Mha or 45%, 133%, 64%, and 122% of those maps, respectively; Supplementary Table 4) if we use more liberal forest criteria to map forest potential (open or closed forest and at least 30% potential tree cover; Supplementary Table 2). By strictly defining forest, requiring multiple lines of evidence, and accounting for fire when mapping forest potential, we limit our constrained reforestation potential to the areas most likely to support dense forest. We thus minimize potential biodiversity and ecosystem services conflicts associated with adding dense tree cover to grassy biomes[11] and avoid fire-maintained and fire-adapted ecosystems. Portions of the constrained potential of other products (8–21%) occur in areas with frequent fire. Because we incorporate fire frequency into our forest potential map, we help to focus on locations where carbon storage is more likely to be durable[46]. We note that reducing fire frequencies in ecosystems with altered fire regimes resulting from invasive species or other causes may be a viable reforestation strategy that would be missed by our approach.

Our methods align with recommendations for right-sizing land-based strategies for carbon dioxide removal by applying sustainability safeguards related to ecological limits, biodiversity, land use competition, and human rights[27]. We acknowledge that our constrained reforestation potential is a substantial reduction from previous estimates and excludes specific reforestation opportunities identified in subnational, national, and regional analyses (e.g., reforestation of marginal croplands with limited food supply ramifications[32]), but our global approach is consistent with do no harm and conservatism principles developed to avoid perverse outcomes, inflated ambition, and misallocated resources for NCS projects[47]. Even without cropland or other precautionary exclusions (Supplementary Table 2), our results demonstrate that reforestation alone cannot meet the forest restoration targets set by IPCC (1000 Mha, as reported by ref.[5]) or in national forest restoration pledges (327 Mha)[48]. Broader approaches to forest restoration that are beyond the scope of this analysis, (e.g., enhancing carbon storage in existing forests[5,8,49]) would be required to achieve those goals.

Our work also illuminates substantial differences in the constrained reforestation potential presented in previously published global maps, which have only 22 Mha in common (Supplementary Fig. 3). Although the global maps use similar methods and produce relatively comparable forest potential areas, decisions regarding which exclusions to apply and the datasets used to represent those exclusions led to highly variable conclusions about the area of constrained reforestation potential. In contrast with existing global maps, we used a broader set of precautionary exclusions, including albedo and wetland exclusions, and used more up-to-date and/or higher resolution LULC products for representing exclusions. We acknowledge uncertainties for our estimates due to our assumptions regarding how forest is defined, which exclusions are appropriate, and which datasets are used to represent key exclusions, such as existing forest and cropland, as demonstrated in our sensitivity analyses. We provide spatial data and code for reproducing our analyses so that users can modify our methods based on their own specific set of assumptions, exclusions,

and/or precautionary principles (see "Data availability" and "Code availability").

Our application of a spatial additionality deduction improves on the non-spatial approach used in the Griscom map[1]. Globally, the rate of forest loss exceeds the rate of forest gain, but there are localized areas where forest regrowth outpaces deforestation (e.g., landscapes with cropland abandonment, Supplementary Fig. 5). If reforestation is intended to provide real and measurable climate change mitigation, it must occur above and beyond baseline forest recovery[47]. However, the best available data of baseline forest recovery remain limited and future research is needed to update the forest loss/gain data[38] to longer time horizons, refine spatial resolution, and disaggregate natural and managed forest loss/gain[50]. We also acknowledge that patterns and rates of forest recovery can rapidly change in response to markets and/or governmental policies or incentives. Other future research to spatially attribute forest responses to these mechanisms would enable more robust additionality evaluation.

We incorporated multiple overlay factors that might help identify specific reforestation opportunities and guide reforestation decision-making (e.g., secure land tenure, biodiversity, and water quality benefits), because additional assumptions and motivations can reduce the area available (Fig. 3a, c). Depending on the factors of interest, there may be a substantial reduction in area relative to the constrained reforestation potential, with an even greater reduction when incorporating multiple factors (Fig. 3a, c). Lack of secure land tenure causes the greatest reduction in area relative to the constrained reforestation potential among the avoiding social conflict scenarios, emphasizing the need for continued progress in formalizing and resolving land rights and title[15] and empowering local land users[51]. Similarly, different measures of land tenure status affect the area identified in the scenario (Supplementary Table 5), highlighting the need for continued development of land tenure information.

We acknowledge that the process of stepping down from a coarse filter, global analysis to identifying real reforestation projects will always require local or national data and incorporate factors that cannot be readily mapped (e.g., social, cultural, or economic considerations that influence landholder decision making[52–55]). Our review of existing maps highlights where finer-scale analyses are available and the types of spatial data those analyses incorporate (Fig. 2 and Supplementary Fig. 1, see "Data availability"). Nonetheless, for most countries, global and regional maps are the best available starting point. Moreover, the maps we produced can be used to highlight areas that merit additional focus or refined analyses. New datasets related to reforestation implementation and opportunity costs[56], soil characteristics[32], wood fuel demand[57], high-resolution poverty measures[58], and carbon durability (e.g., changing fire regimes[59]) present opportunities for additional overlays. Agricultural productivity[60], cropland type[61], and food supply datasets[62], in particular, may help identify marginally productive croplands that could be reforested without impacting local food supply or resulting in conversion of intact ecosystems to meet global food demand[62]. Pastures likely represent a large portion of our constrained reforestation potential, and improved, high-resolution pasture[63] and/or livestock density mapping (as outlined in ref. 64 for 2000) will help clarify the extent of reforestation areas relative to other approaches such as silvopasture.

We demonstrate that global area estimates of the constrained reforestation potential for climate change mitigation are highly variable and are dependent on the criteria and data used for mapping. We used a conservative approach to mapping constrained reforestation potential, which applied biodiversity, ecosystem services, durability, sustainability, and precautionary safeguards and incorporated contemporary, high-resolution LULC datasets. Our resulting area estimates are substantially smaller but less problematic than previous studies, and we demonstrate how additional considerations may further focus where reforestation may be feasible. We argue that applying these conservative considerations to reforestation mapping for policy and practice better supports smart action towards meaningful and equitable climate change mitigation.

## Methods

### Review of existing reforestation maps

We conducted a literature review to identify existing reforestation maps, broadly describe their geographic extent and application of spatial data, and inform our global map (Supplementary Fig. 11). Our main goal was to broadly assess the availability and distribution of recent reforestation maps across scales, not to provide a comprehensive or systematic review. We used the Web of Science Core Collection database to identify an initial pool of 6637 English language journal articles, books, book series, or conference proceedings published between 1 January 2011 and 4 November 2021 based on a search of titles, keywords, and abstracts using the search term ["forest*" and ("map*" or "spatial*") and ("regrow*" or "restor*" or "reforest*")] in a basic search. We then used the abstract screening software Abstrackr[65] to identify articles describing the creation or use of reforestation maps at global, regional, or national scales. We considered a publication to be relevant if it included a spatial assessment of where reforestation could or should potentially occur. We limited our scope to reforestation approaches using natural regeneration or mixed-species planting in terrestrial ecosystems and therefore excluded publications mapping afforestation, plantation or single-species suitability, and wetland or mangrove restoration. We excluded publications focused on describing past patterns of reforestation (i.e., where it has occurred) unless the article used the historical recovery to predict future potential. We also excluded publications focused solely on forest pattern and patch metrics, such as fragmentation, edge, and core area statistics. We stopped screening publications once Abstrackr's learning algorithm predicted a likelihood of relevance less than 0.5, resulting in a pool of 211 potentially relevant publications. We supplemented this list with an additional 20 articles, of which we were previously aware and which did not appear in search results, including studies published in 2022 after our cutoff dates. We read these articles and conducted a final manual screening for relevance. We categorized the remaining 89 publications based on geographical extent (global, regional (i.e., at least three countries, but not global), national (i.e., 100% of one or two countries), or sub-national (i.e., not 100% of any country)); the type of spatial datasets used (32 categories, see Supplementary Table 1); and the application of those spatial datasets as a suitability (i.e., factors used to delineate areas included in the forest potential or maximum/constrained reforestation potential), exclusion (i.e., factors used to remove or mask areas from the forest potential or maximum/constrained reforestation potential), or overlay factor (i.e., factors used for further refinement or prioritization of reforestation potential). Additionally, we associated the extent of each analysis with the corresponding level 0 and 1 administrative unit or units[66] (i.e., the country, countries, or state(s)/province(s) included in the analysis).

### Forest potential mapping

As a first step for generating maps of maximum or constrained reforestation potential, we created a conservative map of forest potential using a process that leverages two existing maps, avoids areas of uncertainty, and incorporates additional spatial data related to fire frequency to address forest potential map critiques. Our forest potential map serves as foundational information for subsequent steps by identifying where tree cover could potentially occur. We identified two raster format forest potential products—the forest potential components of the Bastin[5] and

Walker[8] maps—which met our requirements: (1) they could be used to distinguish between forest and savanna ecosystems, (2) they have higher spatial resolution relative to other forest potential products, and (3) they include spatially-explicit uncertainty data so we could account for pixel-level uncertainty (Supplementary Table 2).

The Bastin map is a continuous prediction of percent potential tree canopy cover at a map resolution of 862 m (meters) produced using a random forest model with climate, topographic, and soils predictors. The product includes an uncertainty layer of the standard deviation of predicted tree cover. The Walker map is a classification of potential aboveground and belowground biomass (AGB and BGB) into closed forest, open forest, and nonwoody systems. The potential AGB map is a continuous prediction at a map resolution of 500 m, also produced using a random forest model with climate, topographic, and soil predictors. Walker et al. combined the potential AGB model with an existing model of root:shoot ratios[67] to estimate potential belowground biomass (BGB), then classified the AGB + BGB values based on bioclimate zone-specific biomass thresholds. Walker et al. produced a pixel-scale uncertainty layer of their AGB prediction as an uncertainty index (UI) representing the range of the 97.5 and 2.5 percentile bounds derived from quantile regression forests divided by the AGB prediction. We re-expressed the Bastin et al. potential tree cover uncertainty as a UI to facilitate comparison with the Walker et al. product at the pixel scale by converting the standard deviation of potential tree cover to a 95% confidence interval, then dividing by the mean potential tree cover prediction.

We created a preliminary composite forest potential map based on a pixel-scale comparison of the Bastin and Walker forest potential and forest potential UI maps after rescaling all products to a 1 km (kilometers) spatial resolution using bilinear resampling. We classified pixels as forest potential if a pixel was mapped as greater than 60% potential tree cover in the Bastin forest potential map, as closed forest in the Walker forest potential map, and with UI values less than 3 in the Bastin and Walker uncertainty maps. This conservative approach thus only maps forest potential to pixels where both the Bastin and Walker forest potential maps agree, and where both products have low uncertainty.

As a final step to map forest potential, we excluded areas that are likely to support fire-dependent grassy ecosystems in places where the climate could also support dense forest[33]. We used a 500 m resolution global monthly burned area dataset derived from MODIS satellite observations[68] to map fire frequency for 2002–2020 by counting the number of burned dates (indicated by MODIS active fire observations and change in burn-sensitive vegetation index values). We rescaled the fire frequency dataset to 1 km using maximum value resampling. To further distinguish between fires in natural ecosystems and fires associated with land clearing and conversion or agricultural practices, we used a 30 m global cropland map agreement dataset for 2022[69] (retaining any pixel identified as cropland in three or more of the six maps evaluated for agreement) and a 30 m resolution global oil palm plantation dataset for 2019[70]. We rescaled the cropland and oil palm plantation products to 1 km using nearest neighbor resampling to match the resolution of the fire frequency map, then removed areas that had 2 or more fires from 2002 to 2020 and that were not cropland or oil palm from the forest potential map.

To evaluate the sensitivity of the forest potential map to our criteria for defining forests, we generated four alternative forest potential maps using identical methods except for the following changes: (1) including areas with frequent fire (instead of excluding those areas), (2) using a 50% potential tree cover threshold in the Bastin forest potential map to define forest (instead of 60%), (3) using a 30% potential tree cover threshold in the Bastin forest potential map and closed forest and open forest in the Walker forest potential map to define forest (instead of 60% and closed forest only), and (4) using a 30% potential tree cover threshold in the Bastin forest potential map to define forest and closed forest and open forest in the Walker forest potential map to define forest (instead of 60% and closed forest only) and including areas with frequent fire (instead of excluding those areas) (Supplementary Table 2).

**Maximum and constrained reforestation potential mapping**

The maximum reforestation potential map shows where it is possible to add forest within the forest potential map, while the constrained reforestation potential map shows where it is possible to add forest within the forest potential map while avoiding perverse outcomes. Our review of spatial datasets used in existing reforestation maps informed our selection of spatial datasets that served as exclusions. We used these datasets to exclude areas from our forest potential map and thus create the maximum and constrained reforestation potential maps. We used implementation exclusions (existing forest, open water, bare ground, and permanent ice/snow) to map maximum reforestation potential, and precautionary exclusions (croplands, built-up lands, peatlands, wetlands, and albedo) to map constrained reforestation potential. Supplementary Table 3 outlines the source spatial datasets and specific pre-processing steps associated with each exclusion; all exclusion datasets are in raster format.

When excluding existing forests to generate the maximum reforestation potential map, we avoided the data critique related to under-mapping existing tree cover in dryland biomes[17] because those systems were already largely absent from our conservative forest potential map. When generating the constrained reforestation potential map, we excluded croplands to address food supply and leakage concerns[20,30,31]. We applied the peatland and wetland precautionary exclusions to protect large soil organic carbon stocks that may be vulnerable to losses following the establishment of trees[71,72]. Additionally, we incorporated albedo as a precautionary exclusion to remove areas where the net climate benefit from reforestation is predicted to be negative due to albedo[26].

As general pre-processing steps for all datasets, we reclassified each dataset to 1 (exclusion) or 0 (not exclusion) and rescaled to 30 m resolution using nearest neighbor resampling. We combined all implementation exclusions into a single raster dataset by summing the individual rasters and setting values greater than 1 to 1. We resampled this dataset to 500 m resolution using pixel averaging to generate an intermediate proportion-excluded raster for subsequent processing steps, then to 1 km resolution, again using pixel averaging, to generate a final proportion-excluded raster. We generated a raster representing maximum reforestation potential by subtracting the proportion excluded raster from 1 for pixels within the forest potential map. We repeated these steps for implementation and precautionary exclusions combined and generated a raster representing constrained reforestation potential by subtracting the proportion excluded raster from 1 for pixels within the forest potential map. The resulting pixel values from these steps indicate the proportion of each pixel where reforestation could occur. To report the area within the maximum or constrained reforestation potential, we multiplied the reforestation potential raster pixel values by their area in hectares.

To evaluate the sensitivity of the constrained reforestation potential to individual datasets used as exclusions, we generated two alternative constrained reforestation potential rasters using identical methods except for the following changes: (1) using a forest/non-forest map derived from radar satellite data[73] instead of a forest extent map derived from multi-spectral satellite data[38] for representing existing forests as an implementation constraint, and (2) using a single product for representing cropland extent[37] instead of a cropland agreement map[69] for representing croplands as a precautionary constraint. Use of radar-based forest extent maps follows recommendations for addressing the limits of forest extent maps in areas with low tree cover[17]. Additionally, we generated a version of the constrained

reforestation potential without using croplands as a precautionary exclusion.

## Overlays and additional scenarios

Similar to our process of identifying exclusions, our review of spatial datasets used in existing reforestation maps informed our selection of overlay data. Overlays include datasets used to represent the feasibility of reforestation within the constrained reforestation potential through additional scenarios as well as reference datasets, such as climate change mitigation and population estimates. These overlays are used to contextualize the constrained reforestation potential and scenario results. Supplementary Table 3 outlines the source, scale, and pre-processing associated with spatial datasets used as overlays.

As general pre-processing steps for all scenario overlays, we converted all source datasets to raster format and set values to 1 to represent features of interest. For the slope and landform datasets used in the water quality scenario, we resampled the 90 m source rasters to 30 m and reprocessed the exclusion datasets to reclassify all pixels outside of moderate slope or floodplain areas as exclusions (see "Maximum and constrained reforestation potential mapping"). Except for the forest neighborhood dataset, we generated all other scenario datasets at 500 m resolution and applied them to the intermediate 500 m proportion excluded raster (see "Maximum and constrained reforestation potential mapping"), reclassifying all pixels outside the scenario overlays as exclusions. We generated forest neighborhood datasets at 1 km resolution and applied them to the 1 km proportion excluded raster (see "Maximum and constrained reforestation potential mapping"), and reclassified all pixels outside the scenario overlay as exclusions. We then generated the final scenario maps and area summaries using methods for maximum and constrained reforestation potential mapping described above.

For the climate change mitigation and population overlays, we reprojected these 1 km datasets to match the 1 km maximum and constrained reforestation potential. We estimated the annual net climate change mitigation for the maximum and constrained reforestation potential and scenarios as $TgCO_2e$ per year by multiplying the maximum and constrained reforestation potential raster pixel values (proportion of each pixel where reforestation could occur) by their area (hectares) by the per-pixel AGB and BGB sequestration potential during the first 30 years of natural regrowth[28] with an offset to account for albedo effects[26] ($TgCO_2e$ per hectare per year). We estimated the total human population directly affected by the maximum and constrained reforestation potential and each scenario (i.e., inhabiting the area) by multiplying the maximum and constrained reforestation potential raster pixel values (proportion of each pixel where reforestation could occur) by a per-pixel estimate of human population in 2020[74].

To evaluate additionality, we summarized the total area of forest loss and forest gain for 2000–2020[36] for one-degree cells, calculated a per-cell loss:gain ratio with maximum value set to 1, and used the loss:gain ratio as a multiplier for the per-cell constrained reforestation potential area and climate change mitigation values. We then summed all cell areas and net mitigation estimates.

We generated four alternative scenarios to evaluate the sensitivity of the secure land tenure scenario to our choice of dataset for representing tenure security. These alternative scenarios are described in detail in Supplementary Table 5.

## Comparison with existing products

We compared agreement and described the maximum extent of the four original global forest potential maps (FLRO, Bastin, Strassburg, and Walker, Supplementary Table 2; we excluded Griscom et al., Cook-Patton et al., and Zheng et al. from comparison because those analyses are derived from FLRO). We reclassified each original map based on the forest definitions applied in the respective publication (Supplementary Table 2). For the FLRO map, we retained pixels classified as potential woodlands, open forests, or closed forests. For the Bastin map, we retained pixels with a potential tree cover greater than 10%. For the Strassburg map, we retained any pixels with a dominant original ecosystem type of forest. For the Walker map, we retained any pixels classified as closed forest or open forest. We then compared the maximum extent and agreement of the four original global constrained reforestation potential maps (FLRO, Griscom, Bastin, and Walker; Supplementary Table 2). We evaluated the Strassburg maximum reforestation potential map separately (described below) and excluded Cook-Patton et al. and Zheng et al. from comparison because those analyses are derived from Griscom. We also reclassified each product according to each publication's definition of constrained reforestation potential (Supplementary Table 2). When publications subdivided reforestation using multiple forest definitions (e.g., by different potential tree canopy thresholds, as in FLRO), we used the broadest definition presented in the publication. For FLRO, we retained pixels classified as wide-scale, mosaic, or remote restoration; for Griscom, we retained pixels identified with reforestation as an NCS; for Bastin, we retained pixels with tree cover restoration greater than 10% and without existing tree cover greater than 10%; and for Walker, we retained pixels mapped with an NCS opportunity as restore/high suitability for forestry-based NCS or restore/low suitability for forestry-based NCS.

To calculate the global area of map agreement, we reclassified all retained pixel values to 1 (otherwise to 0), rescaled the resulting maps to 500 m using nearest neighbor resampling, summed all reclassified and rescaled products, and summarized the global area by agreement count. For each forest potential map, we also calculated the area of agreement (intersection) with our forest potential map. For each constrained reforestation potential map, we calculated the area of agreement (intersection) with our forest potential map, constrained reforestation potential map, implementation exclusions, and precautionary exclusions. For the exclusions used to generate the Bastin and Walker constrained reforestation potential maps, we also calculated the area of agreement (intersection) with our forest potential map, constrained reforestation potential map, implementation exclusions, and precautionary exclusions. For the Strassburg forest restoration map, we identified the maximum reforestation potential by retaining pixels with a restoration opportunity greater than 50% of pixel area and with a dominant original ecosystem type of forest. We calculated the area of agreement of this map with our maximum reforestation potential map and implementation exclusions.

We conducted all spatial data analysis using Google Earth Engine[75], Python version 3.11.8, and the arcpy library, or R version 4.4.1 and the terra library. We conducted all tabular analysis using the tidyverse R library. We generated all figures using ArcGIS Pro version 3.3.1[76] or the ggplot2 and upsetr R libraries. Country boundaries in all map figures are sourced from GADM[66] (https://gadm.org).

## Reporting summary

Further information on research design is available in the Nature Portfolio Reporting Summary linked to this article.

## Data availability

The constrained reforestation potential map can be downloaded and viewed at: https://www.naturebase.org, while archival copies of all spatial and tabular data generated in this study are available on Fig-Share: https://doi.org/10.6084/m9.figshare.27335799. An interactive map viewer with reforestation scenario results by jurisdiction is available at: https://www.reforestationhub.org/global.

## Code availability

Code and instructions for replicating our analyses are available on FigShare: https://doi.org/10.6084/m9.figshare.27335799.

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

## Acknowledgements

K.A.F., E.E.P., D.E.T.H., P.W.E., and S.S.C.-P. were funded by the Bezos Earth Fund. J.W.V. was supported by USDA-NIFA Sustainable Agricultural Systems Grant 2019-68012-29819, USDA-NIFA McIntire-Stennis Project 1016880, and the National Science Foundation under award DEB-1931232. F.F. and P.C. were funded by NASA grant 80NSSC22K1363, and F.F. acknowledges the receipt of a fellowship from the OECD Co-operative Research Programme: Sustainable Agricultural and Food Systems in 2023. S.A. and M.A. were funded by Future Ecosystems for Africa in partnership with Oppenheimer Generations Research and Conservation. M.E.F. was funded by a NASA grant 80NSSC21K0297 and a Bezos Earth Fund grant to the World Resources Institute with a sub-grant to the University of Maryland, Baltimore County. E.C.F. and C.T. were supported by the MIT Climate and Sustainability Consortium and the Government of Portugal through the Portuguese Foundation for International Cooperation in Science, Technology, and Higher Education. N.H. and C.A.W. were funded by a Bezos Earth Fund grant to the Nature Conservancy with a sub-grant to Clark University. We are grateful for the insights and review provided by Christina Kennedy, Johan Oldekop, and David Theobald during the development of our methods, and for cartographic advice provided by Chris Bruce.

## Author contributions

K.A.F., E.E.P., D.E.T.H., and S.C.C.-P. designed the study. J.W.V., F.F., P.C., S.A., M.A., M.E.F., E.C.F., C.T., N.H., C.A.W., and P.W.E. contributed methods. K.A.F. conducted the analyses. K.A.F., E.E.P., and S.C.C.-P. wrote the article. K.A.F., E.E.P., D.E.T.H., J.W.V., F.F., P.C., S.A., M.A., M.E.F., E.C.F., C.T., N.H., C.A.W., P.W.E., and S.C.C.-P. edited the article.

## Competing interests

The authors declare no competing interests.
