## [Transparent Peer Review file · Nature Communications]

Addressing critiques refines global estimates of reforestation potential for climate change mitigation

Corresponding Author: Mr Kurt Fesenmyer

Version 0:

Reviewer comments:

Reviewer #1

(Remarks to the Author)

The subject of this manuscript is very attractive and of a great relevance to the present day. The aim of authors is to refine the potential areas for reforestation worldwide addressing new critiques. To do this the authors provide a literature review of the main reforestation opportunities (n = 89 publications) and then the combinations of seven global, 15 national and 57 sub-national maps adding restriction in order to improve and accurate the potential areas for reforestation management. I really appreciate the incorporation of social aspects and the context of each area for this new approach for reforestation maps. Overall, the authors use appropriate techniques to present the data, and interpret them carefully by combining a wide range of map sources, which gives robustness to the conclusions. In addition, the large amount of supplementary material reinforces the transparency of their results.

However, I have some comments/suggestions:

Specify TgCe/yr units (e.g. total net mitigation potential)

L37: also via direct seeding

L139: add reference to Fig.1

L166: remove the comma "...areas with frequent, natural fire..."

Methods, Review of existing reforestation maps:

- When using WOS, at what level is the search equation entered? by Topic, title, abstract, author keywords...

- Were all types of publications included? (i.e. reviews, original articles, proceedings ...)

- It is highly recommended to register the protocol for systematic reviews (or included in the manuscript as a part of methods section or supplementary material), including all followed steps. Some relevant points in the protocol could be the main research question, the search strategy specifying the search terms and categories from database used, the selection criteria and the reasons for inclusion/exclusion at each step according to PICO items, the PRISMA diagram and data extraction.

L389: why these two maps and not others? In addition, after using two of the seven global maps to generate the new "forest potential map", the results (Results: mapping forest potential section) are compared with 4 of the 7 global maps. Please clarify this.

Due to the large amount of data (datasets, areas, %, ...) it is sometimes a bit confusing to follow. It is not very clear when, how many and why which maps or datasets are used. Please clarify this.

Despite the large number of results in this study, I believe that the authors synthesize the results very well in the discussion. I enjoyed reading it.

(Remarks on code availability)

Reviewer #2

(Remarks to the Author)

Comments on "Addressing critiques refines global estimates of reforestation potential for climate change mitigation"

The comparison and discussion about different maps that identify reforestation potential is well done and useful for fellow researchers, especially those not as immersed in this area as the authors. This alone will make this article a well referenced

science contribution down the road. The authors then add new factors that can be mapped spatially and claim a better, or perhaps, more refined map, as the title indicates. Their refined results suggest just 195 million hectares are suitable for reforestation around the world when cropland, built-up land, wetlands, peatlands, and areas with negative albedo are excluded.

1) A general concern with the whole enterprise of creating ever more refined, or "contextualized", maps is that each new map essentially adds a new "context," which is inevitably something new or different that can either now be mapped, or was ignored by the previous map makers. Use of the word context is itself misleading because it belies a high level of confidence that the "context" of any XX meter pixel is well understood. For example, you have assumed cropland cannot be converted to forests, but the decision to reforest actually is a human decision made locally. I would argue that making that decision for a person who manages the landscape with your mapped cropland and forestland layers is devoid of context rather than "contextualized." If anything, the US Forest Service landowners survey, the thousands of farmer and forestland owner surveys conducted at land grant schools in the US or throughout Europe, Asia, and Latin America, tells us that context is mostly about the incentives (all, not just economic or government) people face locally through social interactions and markets, something not represented in your maps. I don't think this observation reduces the value of your new map, but it would be useful to introduce some caveats to the map making enterprise, including references perhaps to the extensive survey research literature that provides what many social scientists will think of as "context".

2) Why does eliminating croplands make sense from a food security standpoint, especially in places like the United States, Australia, New Zealand, Europe, Brazil, Argentina, and pretty much anywhere in Asia? Food security in those locations has little if anything to do with the number of hectares in crops and far more to do with income. Hertel and others also showed that the intensification margin in these locations is strong, substantially reducing any effects of moving land out of food production in any given location. Furthermore, reforestation would begin on the least productive lands (lowest cost), limiting any food quantity implications. In other places besides these rich countries, food security has more to do with distribution of food than with hectares devoted to food production (per Amartya Sen). Besides, a carbon payment of \$10-\$30 per ton CO₂ on regrowing forests would raise incomes substantially in many cropped locations (assuming the money makes its way to the people who actually manage the land), further eliminating food security concerns in developing countries. Eliminating all cropland is way too broad a brush to use when considering what hectares can be reforested. Besides, there is a fairly active reforestation frontier already in place in most parts of the world, amounting to millions of hectares (see analysis by Richard Houghton's bookkeeping model in various iterations over the years, including their original piece in the early 1980s; Brown and Lugo in the 1990s; Pan et al., 2011; Busch et al., 2019; Harris et al. 2020; etc.).

3) Some of the sensitivity analyses are not clearly explained and do not have strong policy rationale. The water quality context is one example. That scenario seems to further limit pixels to those that meet the base context reforestation criteria (195 Mha) plus newly added water quality criteria. First, what kind of policy would actually encourage this approach? You must have something in mind, but it should be explained, if not in the main text then in the supplement. One issue with such an approach would be additionality (Do US Farm Bill, China conservation payments, European multi-functionality payments, Australian water markets, etc., already pay for actions on these lands?). A more important issue is that this isn't even coming close to finding the right solution to the "optimal" or "best" water and carbon map. It's an artificial construction based on already filtering down to some land that can possibly be reforested, and thus constrained by that. Isn't the right answer to this map an overlay of the best sites for water quality practices and the best sites for reforestation? I don't think that map would look anything like the resulting 71 million hectares you get under this approach. This isn't to say what you have is not useful, but there is virtually no discussion in the paper about how you arrived at this particular approach to add water quality on top of reforestation on top of additionality determine one particular way, etc. etc. For this and other "contexts" analyzed, it would be useful to have more description about the policies you are intending to support with the resulting maps.

4) Explain and show the effect the additionality rule/approach had on the map, i.e., how much land is excluded because it is non-additional? Explain why your approach is an appropriate one for additionality? One issue is why is 2000-2012 even close to a proper measure to use going forward? Brazil was implementing stringent policy to slow deforestation. China paid huge sums to reforest in its sloping lands and other programs. The industrial wood market in many places cratered after 2007, even as demand from China remained strong. Need more justification for why you think this past period is a good predictor going forward, other than the fact that the maps are only available after 2000. A second issue is that you are only actually trying to measure carbon offset additionality. From the perspective of any Annex I country that reports forest fluxes in their UNFCCC, and from the perspective of most NDCs by other countries, this measure of additionality is meaningless since they will count the fluxes by all forests, whether reforested above and beyond a 2000-2012 baseline or not. Suppose a country regulates forests and institutes a law that requires X% of each pixel to be reforested each year, where X% > Y%, and Y% is your calculated additionality deduction (e.g., the proportion of each pixel that reforested from 2000-2012 if I understand what you have done). The country will count the carbon in the full X% in their UNFCCC accounts, or towards their NDC. Yet your additionality constraint suggests they can only count X% - Y%. my recommendation is to be more careful in your discussion of additionality. The additionality constraint used here is specific to selling carbon in an offset market, so your map is specific for organizations like TNC who want to generate additional carbon that can be sold for revenue to fund conservation. However, this is not an additionality constraint that is meaningful for the atmosphere or for policy makers writ large, who are happy to engage in all kinds of non-payment policies to maintain or increase forest uptake.

(Remarks on code availability)

(Remarks to the Author)

The manuscript titled "Addressing critiques refines global estimates of reforestation potential for climate change mitigation" represents a comprehensive study of global opportunities for reforestation as a climate change mitigation strategy. The authors developed new global maps that address key criticisms of previous global maps. They found that the area suitable for reforestation is up to 195 million hectares, 71-92 % less than previous estimates.

The authors conducted a thorough review of existing reforestation maps, identified key critical points, and developed new maps that address them. They used multiple lines of evidence to identify potential forested areas. They also applied a number of exclusions related to implementation constraints and precautionary measures to identify maximum and constrained reforestation potential. Thus, the authors' methodology is comprehensive and justified. The integrated analysis approach included the identification of potential forest areas, maximum and limited reforestation potential, and additional scenarios considering various factors such as human rights, land tenure, livelihood impacts, ecosystem services and state policies. They also conducted sensitivity analyses for different methodological choices. This comprehensive approach offers a more accurate understanding of the potential benefits and constraints associated with the use of reforestation as a climate change mitigation strategy. Previous assessments have been significantly overestimated, which can lead to false expectations and inefficient resource allocation. The authors' more conservative and integrated approach provides a better understanding of where and how much reforestation can be realized, taking into account various environmental, social and political factors. This is important information for policy makers and practitioners designing and implementing climate change strategies.

Thus, the manuscript presents a well-designed and thoughtful approach to remedy the shortcomings of previous works on global reforestation mapping. Nevertheless, there are several areas in which I believe the manuscript should be further developed:

1. While the authors have done some aspects, a fuller discussion of the uncertainties and limitations associated with the different datasets, assumptions, and methodological choices would have been valuable.
2. The authors have effectively conceptualized the results; however, I believe there is an opportunity to provide a more detailed analysis of the practical implications of these new maps. What key circumstances or conditions should be considered when applying these maps at regional or local scales?
3. Also, authors should identify specific areas where further research or data collection could help refine and improve the mapping approach.

(Remarks on code availability)

Reviewer #4

(Remarks to the Author)

General Comments

Many of the global reforestation maps have come under question for their robustness in predicting where forest restoration is feasible. This study tackles some of the major criticisms to produce new maps that are more robust. The study looks at different scenarios where reforestation aims to produce different outcomes, whether that is for ecosystem services and or social and governance issues. The scale of the study is impressive as the authors review 89 reforestation maps. Probably most interesting and important to other scientists, policy makers, and land managers is the finding that their estimates of reforestation cover potential are 71-92% lower than other published estimates. The novelty is also in the fact that the authors use datasets that consider equity issues in mapping (beyond simple biophysical measures that are commonly applied in this type of work).

I support the publication of this manuscript, but there are a few issues that can be addressed before publication. I outline some of the general topics here and specific line comments below.

One of the benefits of these global maps is being able to see where potential reforestation initiatives can be targeted, but also a challenge with these maps is the coarse spatial resolution and difficulty in downscaling these to management-level (often plot-level) metrics (although, of course, a strength of this research is the resolution for a global map). I think this should be mentioned in the discussion because that is also another big critique of the maps that have been published so far. Of course, given the data that are accessible now, getting resolutions at management-level is difficult and is beyond the scope of this paper, but should be at least mentioned.

An impressive piece of this work is that some social aspects of forest restoration were included in the maps. It is a bit unclear to me, however, with the information given, when those datasets were accessed and until what year those datasets cover. Could this type of information be included in S3? For replicability in the future, this would be particularly beneficial.

Specific comments

Manuscript

L71-75. I am glad to see the critiques explicitly stated here and I also wonder if the underlying piece of these is that some people may just not want to reforest. Some of the framing around reforestation and scaling up seems to de-emphasize or completely ignore the autonomy of individuals across a landscape.

L166. How was 'natural' fire defined exactly? It is a bit unclear to me on the tables how a natural fire is or is not distinguishable from a fire started unintentionally by a human. I am wondering too how future fire projections and frequency

were dealt with in excluding areas.

L275-276. I would consider putting the 607 TgCe/yr for first 30 years in more context, perhaps in relation to what that would offset.

Figures

L743, Fig 3. This is a nice figure to conceptualize the study. I think the caption can be improved because, though the figure is helpful, it is not completely intuitive. Describing, in greater detail, the points in Fig3a would be useful. It also takes a moment to see that there are arrows pointing from a to b (in particular). Is there a way to subtly highlight the scenario and connect it to the arrows?

L749, Fig 4. This figure does a nice job of highlighting differences among studies. This is just a stylistic comment, but I wonder two pieces might improve the intuitiveness of the figure. (1) Could a small legend that shows what purple and green represent be added? (2) Could the two last columns and F be highlighted subtly in some way to make it more obvious that they are produced from this study?

(Remarks on code availability)

I had difficulty running the code (a few errors that I could not de-bug) so was not able to fully run the code as is. That being said, I did not find any problems with the way the analysis is done in GEE.

Version 1:

Reviewer comments:

Reviewer #2

(Remarks to the Author)

The authors have addressed my comments adequately.

(Remarks on code availability)

Reviewer #3

(Remarks to the Author)

Dear authors,

Thank you for providing detailed clarifications and revisions based on the comments.

The expanded discussion on the practical aspects of applying the developed maps, including at regional and local scales, increases their utility for planning and implementing climate change mitigation strategies. The additional details make the results more actionable and accessible to a diverse audience.

I agree with the authors' statement: "We believe our updated approach to presenting the sensitivity analyses navigates this challenge by focusing on the methodological and dataset decisions with the largest implications on the results." This approach appropriately balances the need for comprehensive analysis with the clarity of presentation, ensuring that the focus remains on the most impactful methodological choices and dataset considerations.

The additions concerning opportunities for future research are well-conceived. The identified prospects for refining data on pastures, agricultural lands, and economic aspects broaden the potential applications of the findings and pave the way for subsequent studies.

Overall, the revisions significantly enhance the value of the presented work. The responses and changes to the manuscript address the main comments and demonstrate careful consideration in addressing identified limitations.

I support the publication of this article and consider it an important step in advancing the understanding of the mitigation potential of reforestation in addressing climate change.

(Remarks on code availability)

Reviewer #1 comments

Reviewer #1 comment	Response
The subject of this manuscript is very attractive and of a great relevance to the present day. The aim of authors is to refine the potential areas for reforestation worldwide addressing new critiques. To do this the authors provide a literature review of the main reforestation opportunities (n = 89 publications) and then the combinations of seven global, 15 national and 57 sub-national maps adding restriction in order to improve and accurate the potential areas for reforestation management. I really appreciate the incorporation of social aspects and the context of each area for this new approach for reforestation maps. Overall, the authors use appropriate techniques to present the data, and interpret them carefully by combining a wide range of map sources, which gives robustness to the conclusions. In addition, the large amount of supplementary material reinforces the transparency of their results. However, I have some comments/suggestions:	Thank you for your comments and review – we are glad that you see this research as an important and relevant contribution.
Specify TgCe/yr units (e.g. total net mitigation potential)	We have clarified that TgCe/yr refers to total net mitigation potential in the abstract (L11, 14) and throughout the manuscript (L222, 228, 246, 251, 253, 260, 262, 307, 523, 573, 586), Table 1, and supplemental materials.
L37: also via direct seeding	We have added 'direct seeding' to this line (now L39).
L139: add reference to Fig.1	We have added the reference to Fig. 1 at the end of this sentence (now L153).
L166: remove the comma “ ...areas with frequent, natural fire...”	We edited this sentence to address Reviewer #4's comments and the comma is no longer used.
(Review of existing maps) When using WOS, at what level is the search equation entered? by Topic, title, abstract, author keywords...	Thank you for your comments requesting additional details on our review of previously published maps. We have added a number of details that we believe clarify our methods and ensure that the results can be reproduced by others. These edits include additional text in the methods section (L412-445), a PRISMA diagram as a supplemental figure (Fig. S12), additional tables available via Data Availability materials listing the articles identified or retained at each step of the review, and additional code to

Reviewer #1 comment	Response
	enable the replication of Figs 2 and S1 (tables and code available via temporary private link here: https://figshare.com/s/1e6e14b8450580372dbe, public link available upon acceptance). To address this specific comment, we have clarified that our WOS search used the Core Collection database, used a basic search, and used a title, keyword, and abstract search for the keywords ["forest*" and ("map*" or "spatial*") and ("regrow*" or "restor*" or "reforest*")] (L416-420)
(Review of existing maps) Were all types of publications included? (i.e. reviews, original articles, proceedings ...)	We have clarified in the methods that our WOS search focused on English language journal articles, books, book series, or conference proceedings (L416-420).
(Review of existing maps) It is highly recommended to register the protocol for systematic reviews (or included in the manuscript as a part of methods section or supplementary material), including all followed steps. Some relevant points in the protocol could be the main research question, the search strategy specifying the search terms and categories from database used, the selection criteria and the reasons for inclusion/exclusion at each step according to PICO items, the PRISMA diagram and data extraction.	We have updated the methods section (L412-445) and created a PRISMA diagram (Fig. S12) to include additional details on our literature review to ensure that the results can be replicated by others. These details include the main research question, search terms and categories from the database used, selection criteria and reasons for inclusion/exclusion at each step. Additionally, we have provided tables of all publications identified and retained at each review step via Data Availability materials (temporary private link here: https://figshare.com/s/1e6e14b8450580372dbe , public link available upon acceptance).
L389: why these two maps and not others? In addition, after using two of the seven global maps to generate the new “forest potential map”, the results (Results: mapping forest potential section) are compared with 4 of the 7 global maps. Please clarify this.	We have added additional details in the Methods to clarify why we used the Bastin and Walker maps as components of the new ‘forest potential maps’. Specifically: "We identified two raster-format forest potential products – the forest potential components of the Bastin and Walker maps - which met our requirements: (1) they could be used to distinguish between forest and savanna ecosystems, (2) they have higher spatial resolution relative to other forest potential products, and (3) they include spatially-explicit uncertainty data so we could account for pixel-level uncertainty (Table S2)" (L451-455).

Reviewer #1 comment	Response
	Regarding the second comment (comparison of 4 of 7 global maps), we only conducted the comparisons on the unique global products – the 3 products we don't compare are all derived from FLRO, one of the products we do compare. We have added additional clarification throughout the paper to highlight that all comparisons are among the unique global products (L142-144; L155-157; L593-596; L601-605), added Zheng et al. 2022 details to Table S2 so that all 7 global products are described, and added additional references to Table S2 in the methods (L594, 603)
Due to the large amount of data (datasets, areas, %, ...) it is sometimes a bit confusing to follow. It is not very clear when, how many and why which maps or datasets are used. Please clarify this.	Thank you for this comment. We have made a number of edits throughout the document, including the changes recommended in your previous comments (see above), to further clarify when, how many, and why various map products and datasets are used in the analysis.
Despite the large number of results in this study, I believe that the authors synthesize the results very well in the discussion. I enjoyed reading it.	Thank you.

Reviewer #2

Reviewer #2 comment	Response
The comparison and discussion about different maps that identify reforestation potential is well done and useful for fellow researchers, especially those not as immersed in this area as the authors. This alone will make this article a well referenced science contribution down the road. The authors then add new factors that can be mapped spatially and claim a better, or perhaps, more refined map, as the title indicates. Their refined results suggest just 195 million hectares are suitable for reforestation around the world when cropland, built-up land, wetlands, peatlands, and areas with negative albedo are excluded.	Thank you for your comments and review – we are glad that you see this research as a useful contribution.
1) A general concern with the whole enterprise of creating ever more refined, or "contextualized", maps is that each new map essentially adds a new "context," which is inevitably something new or different that can either now be mapped, or was ignored by the previous map makers. Use of the word context is itself misleading because it belies a high level of confidence that the "context" of any XX meter pixel is well understood. For example, you have assumed cropland cannot be converted to forests, but the decision to reforest actually is a human decision made locally. I would argue that making that decision for a person who manages the landscape with your mapped cropland and forestland layers is devoid of context rather than "contextualized." If anything, the US Forest Service landowners survey, the thousands of farmer and forestland owner surveys conducted at land grant schools in the US or throughout Europe, Asia, and	Thank you for this comment – we have removed our use of the term 'context' from the Introduction and have added a sentence to the Discussion that describe how global analyses are used: ("Global restoration mapping efforts, including reforestation maps, are used for estimating the overall magnitude of opportunity, supporting global policy development, and mobilizing resources to places with greater relative opportunity (Wyborn and Evans 2021).", L301-303). Additionally, we added a sentence on the need to evaluate additional factors before implementing restoration: ("We acknowledge that the process of stepping down from a coarse filter, global analysis to identifying real reforestation projects will always require local or national data and incorporate factors that cannot be readily mapped (e.g., social, cultural, or economic considerations that influence land holder decision making (Powlen and Jones 2019, Jones et al. 2020, Toma and Buisson 2022, Godoy et al. 2024))", L379-382). We have added 4 references to this last addition related to landowner motivations for participating in restoration or conservation activities: • Powlen et al. 2019. Identifying the determinants of and barriers to landowner participation in reforestation in Costa Rica. Land Polic. 84, 216–225• Jones et al. 2020. Participation in payments for ecosystem services programs in the Global South: A systematic review. Ecosyst. Serv. 45, 101159

Reviewer #2 comment	Response
Latin America, tells us that context is mostly about the incentives (all, not just economic or government) people face locally through social interactions and markets, something not represented in your maps. I don't think this observation reduces the value of your new map, but it would be useful to introduce some caveats to the map making enterprise, including references perhaps to the extensive survey research literature that provides what many social scientists will think of as "context".	 • Toma and Buisson. 2022. Taking cultural landscapes into account: Implications for scaling up ecological restoration. Land Polic. 120, 106233 • Godoy et al. 2024. Reviewing factors that influence voluntary participation in conservation programs in Latin America. For. Polic. Econ. 169, 103359
2) Why does eliminating croplands make sense from a food security standpoint, especially in places like the United States, Australia, New Zealand, Europe, Brazil, Argentina, and pretty much anywhere in Asia? Food security in those locations has little if anything to do with the number of hectares in crops and far more to do with income. Hertel and others also showed that the intensification margin in these locations is strong, substantially reducing any effects of moving land out of food production in any given location. Furthermore, reforestation would begin on the least productive lands (lowest cost), limiting any food quantity implications. In other places besides these rich countries, food security has more to do with distribution of food than with hectares devoted to food production (per Amartya Sen). Besides, a carbon payment of \$10-\$30 per ton CO2 on regrowing forests would raise incomes substantially in many cropped locations (assuming the money makes its way to the people who actually manage the land), further eliminating food security concerns in developing countries.	Thank you for this comment. We acknowledge the complexity of reforestation of croplands. In the manuscript, our overall objective was to create a global reforestation opportunity map that addresses critiques of previous maps. Inclusion of croplands as areas of opportunity in global reforestation maps has been critiqued (see: Doelman, J. C. & Stehfest, E. The risks of overstating the climate benefits of ecosystem restoration. Nature 609, E1–E3 (2022)). Additionally, we wanted to develop maps that are consistent with NCS Principle 2.2 (see: Ellis, P. W. et al. The principles of natural climate solutions. Nat. Commun. 15, 547 (2024)). Thus, in name of conservativeness, we excluded croplands from our constrained reforestation potential. However, we recognize that future research could improve upon this conservative assumption by identifying marginal croplands available for reforestation, thereby augmenting the reforestation opportunity in ways that adhere to NCS Principle 2.2 (see below). In recognition of the complexity of reforestation on croplands, we also provide constrained reforestation results without cropland exclusion via our sensitivity analysis: (“If we do not use croplands as an exclusion, constrained reforestation potential increases by 31% (to 255 Mha Table S2, Fig. S10c).”, L292-293). We have edited the main manuscript in multiple places to better acknowledge this complexity. First, we have modified the Introduction to more thoroughly describe the critiques of reforestation maps that include croplands and associated caveats: (“Maps that include croplands as reforestable are criticized for not sufficiently considering local

Reviewer #2 comment	Response
Eliminating all cropland is way too broad a brush to use when considering what hectares can be reforested. Besides, there is a fairly active reforestation frontier already in place in most parts of the world, amounting to millions of hectares (see analysis by Richard Houghton's bookkeeping model in various iterations over the years, including their original piece in the early 1980s; Brown and Lugo in the 1990s; Pan et al., 2011; Busch et al., 2019; Harris et al. 2020; etc.).	food security in some regions, changes in food demand due to diet shifts, and/or leakage (i.e., conversion of ecosystems elsewhere for agriculture) (Doelman and Stehfest 2022)", L73-75). Second, we have expanded the description of the cropland exclusion step commonly applied to constrained reforestation potential mapping: ("Many maps also apply safeguards to minimize perverse outcomes and further limit the area where reforestation could occur (hereafter 'constrained reforestation potential'). For example, croplands are commonly removed in this step because reforestation of these lands can affect food security (Fujimori et al. 2022) and/or result in leakage (Schwarze et al. 2002).", L117-120) This last edit now includes additional citations:  • Fujimori, S. et al. Land-based climate change mitigation measures can affect agricultural markets and food security. Nat Food 3, 110–121 (2022). • Schwarze, R., Niles, J. O. & Olander, J. Understanding and managing leakage in forest based greenhouse gas mitigation projects. Philos. Trans. R. Soc. Lond. Ser. A: Math., Phys. Eng. Sci. 360, 1685–1703 (2002). Finally, we have expanded the Discussion to address this issue: ("We acknowledge that our constrained reforestation potential is a substantial reduction from previous estimates and excludes specific reforestation opportunities identified in subnational, national, and regional analyses (e.g., reforestation of marginal croplands with limited food supply ramifications (Cook-Patton et al. 2020)), but our global approach is consistent with 'do no harm' and conservatism principles developed to avoid perverse outcomes, inflated ambition, and misallocated resources for NCS projects (Ellis et al. 2024)." L331-336). Further, we now emphasize the sensitivity of our analyses to our assumptions: ("We acknowledge uncertainties for our estimates due to our assumptions regarding how 'forest' is defined, which exclusions are appropriate, and which datasets are used to represent key exclusions, such as existing forest and cropland, as demonstrated in our sensitivity analyses. We provide spatial data and code for reproducing our analyses so that users can modify methods based on their own specific set of assumptions, exclusions, and/or precautionary principles (see Data Availability, Code Availability)." L348-353).

Reviewer #2 comment	Response
	Further, we now highlight additional opportunities for future research to identify opportunities for reforestation of croplands, especially marginal croplands, using new datasets: (“Agricultural productivity (Folberth et al. 2020), cropland type (See et al. 2023), and food supply datasets (Hoang et al. 2023), in particular, may help identify marginally productive croplands that could be reforested without impacting local food supply or resulting in conversion of intact ecosystems to meet global food demand (Hoang et al. 2023).” L392-394).
3) Some of the sensitivity analyses are not clearly explained and do not have strong policy rationale. The water quality context is one example. That scenario seems to further limit pixels to those that meet the base context reforestation criteria (195 Mha) plus newly added water quality criteria. First, what kind of policy would actually encourage this approach? You must have something in mind, but it should be explained, if not in the main text then in the supplement. One issue with such an approach would be additionality (Do US Farm Bill, China conservation payments, European multi-functionality payments, Australian water markets, etc., already pay for actions on these lands?). A more important issue is that this isn't even coming close to finding the right solution to the "optimal" or "best" water and carbon map. It's an artificial construction based on already filtering down to some land that can possibly be reforested, and thus constrained by that. Isn't the right answer to this map an overlay of the best sites for water quality practices and the best sites for reforestation? I don't think that map would look anything like the resulting 71 million hectares you get under this approach. This isn't to say what you have is not	Thank you for this comment. We have expanded the ‘Description and rationale’ portion of Table 1 to include additional information on motivations for the scenarios. Specifically for the water quality scenario, its Description and Rationale now reads: (“Constrained reforestation potential limited to hillslopes of 20-35% or floodplains. These locations are a focus of local reforestation efforts (e.g., The Nature Conservancy/American Forest Foundation Mississippi Delta Floodplain Restoration Program) due to their low potential for land use change conflict, high water quality benefits (e.g., nutrient retention and soil stabilization (Lan et al. 2020)), and contribution to climate adaptation (e.g., flood mitigation (Dixon et al. 2016, Shannon et al. 2019)).” (Table 1).) Regarding optimization, the scenarios we present are not “best” or “optimized” by choice, rather they provide maps of any location where a particular scenario could be implemented. As we state in our revised Introduction: “Rather than producing a single area estimate or conducting a prioritization exercise, we aimed to provide a “menu of options” that incorporates additional considerations so that decision makers can evaluate how much area is feasible and desirable for their reforestation efforts given their specific circumstances.” (L92-95). We recognize that this could have been clearer (as noted by your first review comment and Reviewer #3 and #4 comments) and we have added content to the Discussion to describe that process and potential uses of our analysis (See response above). Regarding additionality of the scenarios relative to other policies and incentives, this is a great point. While attempting to quantify the additionality of the scenarios is beyond the scope of this analysis, we have added content to the Discussion highlighting these

Reviewer #2 comment	Response
useful, but there is virtually no discussion in the paper about how you arrived at this particular approach to add water quality on top of reforestation on top of additionality determine one particular way, etc. etc. For this and other "contexts" analyzed, it would be useful to have more description about the policies you are intending to support with the resulting maps.	concerns: (“We also acknowledge that patterns and rates of forest recovery can rapidly change in response to markets and/or governmental policies or incentives. Other future research to spatially attribute forest responses to these mechanisms would enable more robust additionality evaluation.” L362-365). (As a side note, we are currently conducting separate research that uses a global SWAT model (https://swat.tamu.edu/) to evaluate reforestation effects on water quality and quantity and more directly address the reviewer’s great comment about optimal placement of reforestation for both water and reforestation benefits.)
4) Explain and show the effect the additionality rule/approach had on the map, i.e., how much land is excluded because it is non-additional? Explain why your approach is an appropriate one for additionality? One issue is why is 2000-2012 even close to a proper measure to use going forward? Brazil was implementing stringent policy to slow deforestation. China paid huge sums to reforest in its sloping lands and other programs. The industrial wood market in many places cratered after 2007, even as demand from China remained strong. Need more justification for why you think this past period is a good predictor going forward, other than the fact that the maps are only available after 2000. A second issue is that you are only actually trying to measure carbon offset additionality. From the perspective of any Annex I country that reports forest fluxes in their UNFCCC, and from the perspective of most NDCs by other countries, this measure of additionality is meaningless since they will count the fluxes by all forests, whether reforested above and beyond a 2000-2012 baseline or not. Suppose a country regulates forests and institutes a law that requires X% of each pixel to be	Thank you for this important comment. You are absolutely correct that it is difficult to address additionality at the global level, but in contrast to most mapping efforts we felt it was important to include some element of additionality to adhere to NCS Principle 3.1 (Ellis et al 2024) and avoid presenting inflated estimates. We have updated the analysis to use 2000-2020 forest loss and forest gain (Potapov et al. 2021) to more accurately account for additionality – we believe this longer time period is more robust for characterizing long term trends and less susceptible to changes that occur at shorter time periods (e.g., political administrations, market conditions, etc. as the reviewer describes). However, the reviewer makes an excellent point. We recognize that the 2000-2020 time period may not be reflective of rapid changes in forest recover, nor future trends, and we now acknowledge in the text that our approach is a simple framework and opportunity for future analysis and improvement: (“... the best-available data of baseline forest recovery remain limited and future research is needed to update the forest loss/gain data (Potapov et al. 2021) to longer time horizons, refine spatial resolution, and disaggregate natural and managed forest loss/gain (Jung et al. 2023). We also acknowledge that patterns and rates of forest recovery can rapidly change in response to markets and/or governmental policies or incentives. Other future research to spatially attribute forest responses to these mechanisms would enable more robust additionality evaluation.” L359-365). We describe the effect of this updated additionality deduction on the constrained reforestation potential: (“... we approximate that reforestation activities would likely be

Reviewer #2 comment	Response
reforested each year, where $X% > Y%$, and $Y%$ is your calculated additionality deduction (e.g., the proportion of each pixel that reforested from 2000-2012 if I understand what you have done). The country will count the carbon in the full $X%$ in their UNFCCC accounts, or towards their NDC. Yet your additionality constraint suggests they can only count $X% - Y%$. my recommendation is to be more careful in your discussion of additionality. The additionality constraint used here is specific to selling carbon in an offset market, so your map is specific for organizations like TNC who want to generate additional carbon that can be sold for revenue to fund conservation. However, this is not an additionality constraint that is meaningful for the atmosphere or for policy makers writ large, who are happy to engage in all kinds of non-payment policies to maintain or increase forest uptake.	additional within 169 Mha (88.7%) of the constrained reforestation potential and achieve 569 TgCe/yr (93.7%) of net climate benefit.” L226-228). Figure S5, now updated with the 2000-2020 data, shows the ratio of forest loss:gain by one-degree cell. These values are used as multiplier to adjust the constrained reforestation potential area and mitigation by additionality (see Methods L583-586). We have updated the description of Figure S5 to highlight that these values can be interpreted to show the proportion of area or mitigation where reforestation may be additional. We disagree that additionality is not a meaningful constraint for policy makers and for the atmosphere. Additionality is not a concept constrained to the voluntary carbon (“offset”) market, but a base principle of any conservation action (Maron, M., Rhodes, J.R. and Gibbons, P., 2013. Calculating the benefit of conservation actions. Conservation letters, 6(5), pp.359-367), a necessary condition of IPCC scientific assessments (definitions explicitly “exclude natural CO2 uptake not directly caused by human activities.” https://www.ipcc.ch/report/ar6/wg1/downloads/report/IPCC_AR6_WGI_AnnexVII.pdf), and core principle of natural climate solutions in (Principle 3.1 in Ellis et al. 2024). Interestingly, note that this very issue of additionality in natural removals accounting surfaced in a recent paper in Nature (Allen, M.R., Frame, D.J., Friedlingstein, P., Gillett, N.P., Grassi, G., Gregory, J.M., Hare, W., House, J., Huntingford, C., Jenkins, S. and Jones, C.D., 2024. Geological Net Zero and the need for disaggregated accounting for carbon sinks. Nature, pp.1-3) rightly critiquing the claiming of “passive removals” in Net Zero accounting.

Reviewer #3

Reviewer #3 comment	Response
The manuscript titled "Addressing critiques refines global estimates of reforestation potential for climate change mitigation" represents a comprehensive study of global opportunities for reforestation as a climate change mitigation strategy. The authors developed new global maps that address key criticisms of previous global maps. They found that the area suitable for reforestation is up to 195 million hectares, 71-92 % less than previous estimates. The authors conducted a thorough review of existing reforestation maps, identified key critical points, and developed new maps that address them. They used multiple lines of evidence to identify potential forested areas. They also applied a number of exclusions related to implementation constraints and precautionary measures to identify maximum and constrained reforestation potential. Thus, the authors' methodology is comprehensive and justified. The integrated analysis approach included the identification of potential forest areas, maximum and limited reforestation potential, and additional scenarios considering various factors such as human rights, land tenure, livelihood impacts, ecosystem services and state policies. They also conducted sensitivity analyses for different methodological choices. This comprehensive approach offers a more accurate understanding of the potential benefits and constraints associated with the use of reforestation as a climate change mitigation strategy. Previous assessments have been significantly overestimated, which can lead to false expectations and inefficient resource allocation. The authors' more conservative and integrated approach provides a better understanding of where and how much reforestation can be realized, taking into account various environmental, social and political factors. This is	Thank you for your comments and review – we are glad that you see this research as a comprehensive and important contribution.

Reviewer #3 comment	Response
important information for policy makers and practitioners designing and implementing climate change strategies. Thus, the manuscript presents a well-designed and thoughtful approach to remedy the shortcomings of previous works on global reforestation mapping. Nevertheless, there are several areas in which I believe the manuscript should be further developed:	
While the authors have done some aspects, a fuller discussion of the uncertainties and limitations associated with the different datasets, assumptions, and methodological choices would have been valuable.	Thank you for this helpful comment – we agree it is important for readers to understand the implications of various methodological and dataset decisions. Accordingly, we present 4 alternative approaches to forest potential mapping based on different assumptions/methods related to how “forest” is defined. This includes an additional alternative approach from our original submission: (“Using liberal criteria to define forest (>30% potential tree cover and lower biomass thresholds) and removing areas with frequent fire substantially increases forest potential (5,095 Mha, an increase of 127%) and constrained reforestation potential (845 Mha, an increase of 333%; Table S2).” L283-286). Additionally, we present 1 alternative approach to constrained reforestation potential mapping based on a different assumption (no exclusion of croplands), 2 alternatives to constrained reforestation mapping based on the use of different datasets (different layers for representing forest and cropland exclusions, the 2 largest exclusions by area), and 4 alternative approaches to land tenure mapping based on the use of different data. In addition to presenting these sensitivity tests in the Results (see L280-297), Discussion (see L372-377) and Supplementary Tables S2, S5, we have now updated our discussion of these sensitivity analyses to more explicitly acknowledge the uncertainties associated with our assumptions: (“We acknowledge uncertainties for our estimates due to our assumptions regarding how ‘forest’ is defined, which exclusions are appropriate, and which datasets are used to represent key exclusions, such as existing forest and cropland, as demonstrated in our sensitivity analyses. We provide spatial data and code for reproducing our analyses so that users can modify methods

Reviewer #3 comment	Response
	based on their own specific set of assumptions, exclusions, and/or precautionary principles (see Data Availability, Code Availability).” L349-353). We believe it is beyond the scope of the study to expand the sensitivity analyses any further. Another reviewer commented on the challenge of presenting too many different results and we believe our updated approach to presenting the sensitivity analyses navigates this challenge by focusing on the methodological and dataset decisions with the largest implications on the results.
The authors have effectively conceptualized the results; however, I believe there is an opportunity to provide a more detailed analysis of the practical implications of these new maps. What key circumstances or conditions should be considered when applying these maps at regional or local scales?	Thank you for this comment. In response to this comment and similar comments from Reviewer #2, we have expanded the Discussion to further describe the uses of global map analyses (L301-303, L379-383) and the additional information that could be considered when stepping down from a global analysis to actual reforestation projects (L388-397).
Also, authors should identify specific areas where further research or data collection could help refine and improve the mapping approach.	Thank you for this comment. In response to this comment and similar comments from other reviewers, we have expanded the Discussion to provide additional examples of analyses and data development that could be applied to the methods we present and further refine our analyses: (“We also acknowledge that patterns and rates of forest recovery can rapidly change in response to markets and/or governmental policies or incentives. Other future research to spatially attribute forest responses to these mechanisms would enable more robust additionality evaluation.” L362-365). Additionally: (“...the maps we produced can be used to highlight areas that merit additional focus or refined analyses. New datasets related to reforestation implementation and opportunity costs (Zeng et al. 2020), soil characteristics (Cook-Patton et al. 2020), wood fuel demand (Bailis et al. 2015), high-resolution poverty measures (Choksi et al. 2023), and carbon durability (e.g., changing fire regimes (Sayedi et al. 2024)) present opportunities for additional overlays. Agricultural productivity (Folberth et al. 2020), cropland type (See et al. 2023), and food supply datasets (Hoang et al. 2023), in particular, may help identify marginally productive croplands that could be reforested without impacting local food supply or resulting in conversion of intact ecosystems to meet global food demand (Hoang et al. 2023). Pastures likely represent

Reviewer #3 comment	Response
	a large portion of our constrained reforestation potential, and improved, high-resolution pasture (Parente et al. 2024) and/or livestock density mapping (as outlined in ref. (Ramankutty et al. 2008) for 2000) will help clarify the extent of reforestation areas relative to other approaches such as silvopasture." L389-397)

Reviewer #4

Reviewer #4 comment	Response
Many of the global reforestation maps have come under question for their robustness in predicting where forest restoration is feasible. This study tackles some of the major criticisms to produce new maps that are more robust. The study looks at different scenarios where reforestation aims to produce different outcomes, whether that is for ecosystem services and or social and governance issues. The scale of the study is impressive as the authors review 89 reforestation maps. Probably most interesting and important to other scientists, policy makers, and land managers is the finding that their estimates of reforestation cover potential are 71-92% lower than other published estimates. The novelty is also in the fact that the authors use datasets that consider equity issues in mapping (beyond simple biophysical measures that are commonly applied in this type of work). I support the publication of this manuscript, but there are a few issues that can be addressed before publication. I outline some of the general topics here and specific line comments below.	Thank you for your comments and review – we are glad that you see this research as a novel and important contribution.
One of the benefits of these global maps is being able to see where potential reforestation initiatives can be targeted, but also a challenge with these maps is the coarse spatial resolution and difficulty in downscaling these to management-level (often plot-level) metrics (although, of course, a strength of this research is the resolution for a global map). I think this should be mentioned in the discussion because that is also another big critique of the maps that have been published so far. Of course, given the data that are accessible now, getting resolutions at management-level is difficult and is beyond the scope of this paper, but should be at least mentioned.	Thank you for this comment, which is consistent with Reviewers #2 and #3. We have added additional text to the Discussion on how the use cases for global maps: (“Global restoration mapping efforts, including reforestation maps, are used for estimating the overall magnitude of opportunity, supporting global policy development, and mobilizing resources to places with greater relative opportunity (Wyborn and Evans 2021)” L301-303) and how additional spatial and non-spatial information can be used to identify reforestation projects: (“We acknowledge that the process of stepping down from a coarse filter, global analysis to identifying real reforestation projects will always require local or national data and incorporate factors that cannot be readily mapped (e.g., social, cultural, or economic considerations that influence land holder decision making (Powlen and Jones 2019, Jones et al. 2020, Toma and Buisson 2022, Godoy et al. 2024). Our review of existing maps highlights where

Reviewer #4 comment	Response
	finer-scale analyses are available and the types of spatial data those analyses incorporate (Figs. 2, S1, Supplemental Materials). Nonetheless, for most countries, global and regional maps are the best available starting point.” L379-387, with added citations).
An impressive piece of this work is that some social aspects of forest restoration were included in the maps. It is a bit unclear to me, however, with the information given, when those datasets were accessed and until what year those datasets cover. Could this type of information be included in S3? For replicability in the future, this would be particularly beneficial.	Thank you for this comment- we have updated Table S3 with additional information on the year(s) the datasets cover.
L71-75. I am glad to see the critiques explicitly stated here and I also wonder if the underlying piece of these is that some people may just not want to reforest. Some of the framing around reforestation and scaling up seems to de-emphasize or completely ignore the autonomy of individuals across a landscape.	Thank you for this comment – related to your comment above regarding stepping from global analysis to actual, on-the-ground reforestation projects, we have added additional text in the Discussion outlining the use case for global map products (L301-303) and how local and national spatial data can help refine the identifying areas with reforestation opportunity (L379-397) highlighting additional non-spatial data related to individual land holder decision making that influence actual project implementation: (“We acknowledge that the process of stepping down from a coarse filter, global analysis to identifying real reforestation projects will always require local or national data and incorporate factors that cannot be readily mapped (e.g., social, cultural, or economic considerations that influence land holder decision making(Powlen and Jones 2019, Jones et al. 2020, Toma and Buisson 2022, Godoy et al. 2024).” L372-375, with citations).
L166. How was ‘natural’ fire defined exactly? It is a bit unclear to me on the tables how a natural fire is or is not distinguishable from a fire started unintentionally by a human. I am wondering too how future fire projections and frequency were dealt with in excluding areas.	We have rewritten this sentence to remove the term ‘natural’: (“Finally, because the Bastin and Walker models overestimate forest potential in ecosystems with frequent fires (Veldman et al. 2019), we eliminated areas with two or more non-cropland fires during 2002-2022, under the assumption that fire frequencies of at least two per decade can substantially limit tree density (see Methods).” L184-187). Additionally, we have updated the Methods to further clarify our rationale for excluding cropland fires: (“To further distinguish between fires in natural ecosystems and fires associated with land clearing and conversion or

Reviewer #4 comment	Response
	agricultural practices, we used a 30 m global cropland map agreement dataset for 2022 (Tubiello et al. 2023) (retaining any pixel identified as cropland in 3 or more of the 6 maps evaluated for agreement) and a 30 m resolution global oil palm plantation dataset for 2019 (Descals et al. 2021). We rescaled the cropland and oil palm plantation products to 1km using nearest neighbor resampling to match the resolution of the fire frequency map, then removed areas which had 2 or more fires from 2002 - 2020 and that were not cropland or oil palm from the forest potential map.” L486-492). We did not include future fire regime projections as an overlay in this study, but now include future fire regime changes in our Discussion list of example datasets that could be used as additional overlays (L391).
L275-276. I would consider putting the 607 TgCe/yr for first 30 years in more context, perhaps in relation to what that would offset.	We have related this value to global fossil fuel and land use change emissions in the Discussion: “roughly 5% of the sum of global fossil fuel and land use change emissions in 2022” (L307-308)
L743, Fig 3. This is a nice figure to conceptualize the study. I think the caption can be improved because, though the figure is helpful, it is not completely intuitive. Describing, in greater detail, the points in Fig3a would be useful. It also takes a moment to see that there are arrows pointing from a to b (in particular). Is there a way to subtly highlight the scenario and connect it to the arrows?	Thank you for this comment. We edited the figure description to clarify the meaning of the points below the bar graph. Additionally, we have added boxes to highlight the connection between the select scenarios represented in the maps.
L749, Fig 4. This figure is does a nice job of highlighting differences among studies. This is just a stylistic comment, but I wonder two pieces might improve the intuitiveness of the figure. (1) Could a small legend that shows what purple and green represent be added? (2) Could the two last columns and F be highlighted subtly in some way to make it more obvious that they are produced from this study?	Thank you for this comment. We have added a legend to the figure and bolded the outputs of this study in the table. To further simplify the figure, we are now only presenting a comparison of the constrained reforestation potential maps (green) and not the maximum reforestation potential maps (purple).
I had difficulty running the code (a few errors that I could not debug) so was not able to fully run the code as is. That being said, I	We have thoroughly revised our code and data availability to address these concerns. We have created a readme document to guide readers through the process of replicating our entire analysis, updated our code

Reviewer #4 comment	Response
did not find any problems with the way the analysis is done in GEE.	for clarity and consistency, provided additional intermediate data, and deposited our code in a public online repository: (temporary private link: https://figshare.com/s/1e6e14b8450580372dbe with public link created pending acceptance: https://doi.org/10.6084/m9.figshare.27335799)

References

- Bailis, R., R. Drigo, A. Ghilardi, and O. Masera. 2015. The carbon footprint of traditional woodfuels. *Nature Climate Change* 5:266–272.
- Choksi, P., A. Agrawal, I. Bialy, R. Chaturvedi, K. F. Davis, S. Dhyani, F. Fleischman, J. Lechner, H. Nagendra, V. Srinivasan, and R. DeFries. 2023. Combining socioeconomic and biophysical data to identify people-centric restoration opportunities. *npj Biodiversity* 2:7.
- Cook-Patton, S. C., T. Gopalakrishna, A. Daigneault, S. M. Leavitt, J. Platt, S. M. Scull, O. Amarjargal, P. W. Ellis, B. W. Griscom, J. L. McGuire, S. M. Yeo, and J. E. Fargione. 2020. Lower cost and more feasible options to restore forest cover in the contiguous United States for climate mitigation. *One Earth* 3:739–752.
- Descals, A., S. Wich, E. Meijaard, D. L. A. Gaveau, S. Peedell, and Z. Szantoi. 2021. High-resolution global map of smallholder and industrial closed-canopy oil palm plantations. *Earth System Science Data* 13:1211–1231.
- Dixon, S. J., D. A. Sear, N. A. Odoni, T. Sykes, and S. N. Lane. 2016. The effects of river restoration on catchment scale flood risk and flood hydrology. *Earth Surface Processes and Landforms* 41:997–1008.
- Doelman, J. C., and E. Stehfest. 2022. The risks of overstating the climate benefits of ecosystem restoration. *Nature* 609:E1–E3.
- Ellis, P. W., A. M. Page, S. Wood, J. Fargione, Y. J. Masuda, V. C. Denney, C. Moore, T. Kroeger, B. Griscom, J. Sanderman, T. Atleo, R. Cortez, S. Leavitt, and S. C. Cook-Patton. 2024. The principles of natural climate solutions. *Nature Communications* 15:547.
- Folberth, C., N. Khabarov, J. Balkovič, R. Skalský, P. Visconti, P. Ciais, I. A. Janssens, J. Peñuelas, and M. Obersteiner. 2020. The global cropland-sparing potential of high-yield farming. *Nature Sustainability* 3:281–289.
- Fujimori, S., W. Wu, J. Doelman, S. Frank, J. Hristov, P. Kyle, R. Sands, W.-J. van Zeist, P. Havlik, I. P. Domínguez, A. Sahoo, E. Stehfest, A. Tabeau, H. Valin, H. van Meijl, T. Hasegawa, and K. Takahashi. 2022. Land-based climate change mitigation measures can affect agricultural markets and food security. *Nature Food* 3:110–121.
- Godoy, C. C. N., F. C. Speroni, M. Nuñez-Regueiro, and L. O. Girardin. 2024. Reviewing factors that influence voluntary participation in conservation programs in Latin America. *Forest Policy and Economics* 169:103359.

- Hoang, N. T., O. Taherzadeh, H. Ohashi, Y. Yonekura, S. Nishijima, M. Yamabe, T. Matsui, H. Matsuda, D. Moran, and K. Kanemoto. 2023. Mapping potential conflicts between global agriculture and terrestrial conservation. *Proceedings of the National Academy of Sciences* 120:e2208376120.
- Jones, K. W., K. Powlen, R. Roberts, and X. Shinbrot. 2020. Participation in payments for ecosystem services programs in the Global South: A systematic review. *Ecosystem Services* 45:101159.
- Jung, M., M. Lesiv, E. Warren-Thomas, D. Shchepashchenko, L. See, and S. Fritz. 2023. The importance of capturing management in forest restoration targets. *Nature Sustainability* 6:1321–1325.
- Lan, H., D. Wang, S. He, Y. Fang, W. Chen, P. Zhao, and Y. Qi. 2020. Experimental study on the effects of tree planting on slope stability. *Landslides* 17:1021–1035.
- Parente, L., L. Sloat, V. Mesquita, D. Consoli, R. Stanimirova, T. Hengl, C. Bonannella, N. Teles, I. Wheeler, M. Hunter, S. Ehrmann, L. Ferreira, A. P. Mattos, B. Oliveira, C. Meyer, M. Şahin, M. Witjes, S. Fritz, Z. Malek, and F. Stolle. 2024. Annual 30-m maps of global grassland class and extent (2000–2022) based on spatiotemporal Machine Learning. *Scientific Data* 11:1303.
- Potapov, P., X. Li, A. Hernandez-Serna, A. Tyukavina, M. C. Hansen, A. Kommareddy, A. Pickens, S. Turubanova, H. Tang, C. E. Silva, J. Armston, R. Dubayah, J. B. Blair, and M. Hofton. 2021. Mapping global forest canopy height through integration of GEDI and Landsat data. *Remote Sensing of Environment* 253:112165.
- Powlen, K. A., and K. W. Jones. 2019. Identifying the determinants of and barriers to landowner participation in reforestation in Costa Rica. *Land Use Policy* 84:216–225.
- Ramankutty, N., A. T. Evan, C. Monfreda, and J. A. Foley. 2008. Farming the planet: 1. Geographic distribution of global agricultural lands in the year 2000. *Global Biogeochemical Cycles* 22:n/a-n/a.
- Sayedi, S. S., B. W. Abbott, B. Vannière, B. Leys, D. Colombaroli, G. G. Romera, M. Słowiński, J. C. Aleman, O. Blarquez, A. Feurdean, K. Brown, T. Aakala, T. Alenius, K. Allen, M. Andric, Y. Bergeron, S. Biagioni, R. Bradshaw, L. Bremond, E. Brisset, J. Brooks, S. O. Brugger, T. Brussel, H. Cadd, E. Cagliero, C. Carcaillet, V. Carter, F. X. Catry, A. Champreux, E. Chaste, R. D. Chavardès, M. Chipman, M. Conedera, S. Connor, M. Constantine, C. C. Mustaphi, A. N. Dabengwa, W. Daniels, E. D. Boer, E. Dietze, J. Estrany, P. Fernandes, W. Finsinger, S. G. A. Flantua, P. Fox-Hughes, D. M. Gaboriau, E. M. Gayo, Martin. P. Girardin, J. Glenn, R. Glückler, C. González-Arango, M. Groves, D. S. Hamilton, R. J. Hamilton, S. Hantson, K. A. Hapsari, M. Hardiman, D. Hawthorne, K. Hoffman, J. Inoue, A. T. Karp, P. Krebs, C. Kulkarni, N. Kuosmanen, T. Lacourse, M.-P. Ledru, M. Lestienne, C. Long, J. A. López-Sáez, N. Loughlin, M. Niklasson, J. Madrigal, S. Y. Maezumi, K. Marcisz, M. Mariani, D. McWethy, G.

Meyer, C. Molinari, E. Montoya, S. Mooney, C. Morales-Molino, J. Morris, P. Moss, I. Oliveras, J. M. Pereira, G. B. Pezzatti, N. Pickarski, R. Pini, E. Rehn, C. C. Remy, J. Revelles, D. Rius, V. Robin, Y. Ruan, N. Rudaya, J. Russell-Smith, H. Seppä, L. Shumilovskikh, W. T. Sommers, Ç. Tavşanoğlu, C. Umbanhowar, E. Urquiaga, D. Urrego, R. S. Vachula, T. Wallenius, C. You, and A.-L. Daniau. 2024. Assessing changes in global fire regimes. *Fire Ecology* 20:18.

Schwarze, R., J. O. Nilsson, and J. Olander. 2002. Understanding and managing leakage in forest-based greenhouse gas mitigation projects. *Philosophical Transactions of the Royal Society of London. Series A: Mathematical, Physical and Engineering Sciences* 360:1685–1703.

See, L., S. Gilliams, G. Conchedda, J. Degerickx, K. V. Tricht, S. Fritz, M. Lesiv, J. C. L. Bayas, J. Rosero, F. N. Tubiello, and Z. Szantoi. 2023. Dynamic global-scale crop and irrigation monitoring. *Nature Food* 4:736–737.

Shannon, P. D., C. W. Swanston, M. K. Janowiak, S. D. Handler, K. M. Schmitt, L. A. Brandt, P. R. Butler-Leopold, and T. Ontl. 2019. Adaptation strategies and approaches for forested watersheds. *Climate Services* 13:51–64.

Toma, T. S. P., and E. Buisson. 2022. Taking cultural landscapes into account: Implications for scaling up ecological restoration. *Land Use Policy* 120:106233.

Tubiello, F. N., G. Conchedda, L. Casse, H. Pengyu, C. Zhongxin, G. D. Santis, S. Fritz, and D. Muchoney. 2023. Measuring the world's cropland area. *Nature Food*:1–3.

Veldman, J. W., J. C. Aleman, S. T. Alvarado, T. M. Anderson, S. Archibald, W. J. Bond, T. W. Boutton, N. Buchmann, E. Buisson, J. G. Canadell, M. de S. Dechoum, M. H. Diaz-Toribio, G. Durigan, J. J. Ewel, G. W. Fernandes, A. Fidelis, F. Fleischman, S. P. Good, D. M. Griffith, J.-M. Hermann, W. A. Hoffmann, S. L. Stradic, C. E. R. Lehmann, G. Mahy, A. N. Nerlekar, J. B. Nippert, R. F. Noss, C. P. Osborne, G. E. Overbeck, C. L. Parr, J. G. Pausas, R. T. Pennington, M. P. Perring, F. E. Putz, J. Ratnam, M. Sankaran, I. B. Schmidt, C. B. Schmitt, F. A. O. Silveira, A. C. Staver, N. Stevens, C. J. Still, C. A. E. Strömberg, V. M. Temperton, J. M. Varner, and N. P. Zaloumis. 2019. Comment on “The global tree restoration potential.” *Science* 366.

Wyborn, C., and M. C. Evans. 2021. Conservation needs to break free from global priority mapping. *Nature Ecology & Evolution* 5:1322–1324.

Zeng, Y., T. V. Sarira, L. R. Carrasco, K. Y. Chong, D. A. Friess, J. S. H. Lee, P. Taillardat, T. A. Worthington, Y. Zhang, and L. P. Koh. 2020. Economic and social constraints on reforestation for climate mitigation in Southeast Asia. *Nature Climate Change* 10:842–844.

Reviewer #2 (Remarks to the Author):

Reviewer # 2 comment	Response
The authors have addressed my comments adequately.	Thank you for your comments and review.

Reviewer #3 (Remarks to the Author):

Reviewer # 3 comment	Response
Thank you for providing detailed clarifications and revisions based on the comments. The expanded discussion on the practical aspects of applying the developed maps, including at regional and local scales, increases their utility for planning and implementing climate change mitigation strategies. The additional details make the results more actionable and accessible to a diverse audience. I agree with the authors' statement: "We believe our updated approach to presenting the sensitivity analyses navigates this challenge by focusing on the methodological and dataset decisions with the largest implications on the results." This approach appropriately balances the need for comprehensive analysis with the clarity of presentation, ensuring that the focus remains on the most impactful methodological choices and dataset considerations. The additions concerning opportunities for future research are well-conceived. The identified prospects for refining data on pastures, agricultural lands, and economic aspects broaden the potential applications of the findings and pave the way for subsequent studies. Overall, the revisions significantly enhance the value of the presented work. The responses and changes to the manuscript address the main comments and demonstrate careful consideration in addressing identified limitations. I support the publication of this article and consider it an important step in advancing the	Thank you for your comments and review.

understanding of the mitigation potential of reforestation in addressing climate change.	
--	--